# Pragmatic representations of self- and others' action in the monkey putamen

Cristina Rotunno[1], Matilde Reni [1], Carolina Giulia Ferroni [1,2], Ebrahim Ismaiel [1], Gemma Ballestrazzi[1], Elena Borra [1], Monica Maranesi[1,3] & Luca Bonini [1,3] ✉

Social coordination in primates relies on parieto-frontal networks encoding self- and others' actions. These areas send convergent projections to the putamen, but its role in representing self- and others' actions remains unknown. We recorded neuronal activity from anatomically characterized putamen regions of two male macaques during a Mutual Action Task (MAT), where a monkey and a human took turns grasping a multi-affordance object based on sensory cues. Cortico-striatal synaptic input, indexed by local field potentials, mirrored known cortical dynamics during sensory instructions and movement, while single neurons selectively encoded the monkey's action, the human's action, or both. Grip type was encoded only during the monkey's trials. Viewing the partner's action was neither necessary nor sufficient, as neurons fired even when it occurred in darkness but not when viewed through a transparent barrier. Thus, the possibility for actual interaction characterizes the pragmatic role of the putamen in gating cortical representations of self- and other's actions in social contexts.

Coordinating one's own actions with those of others is a fundamental skill in both humans and other nonhuman primates. To this end, we draw on contextual cues, physical objects, and other's observed behavior to decide whether to act and which action to perform. Importantly, such a variety of physical[1] and social[2] stimuli affords a large set of potential motor actions, which can ultimately be selected and converted into overt motor responses, or inhibited when inappropriate in the current context.

To date, neuroscientific research on these abilities has been limited to the cerebral cortex, where several parietal and frontal regions host neuronal mechanisms underlying visuomotor transformations for grasping objects[3–6] and planning motor responses to others' observed actions during social interactions[2,7,8]. Indeed, neurons with partially mixed selectivity for graspable objects, self, and others' actions have been reported in the inferior parietal[9], ventral[10], and mesial[11,12] premotor cortices, as well as in the prefrontal cortex[13]. These regions form a cortical network where different sensory information is remapped onto a motor code, affording possible motor responses to perceived stimuli[14]. This latter conclusion relies on evidence that the visual responses of neurons in these areas are down-modulated or completely abolished when a transparent barrier is interposed between the observer and the observed object[6,9,10] or another's action[7,15]. Since a barrier does not alter the visual features of the stimulus but selectively eliminates the possibility of interacting with it, these findings demonstrate the pragmatic nature of the visual responses in the cortical grasping network, that is, their role in affording object-directed actions as well as motor responses to others during social interactions[14].

Recent neuroanatomical studies have shown that all the aforementioned parietal, premotor, and prefrontal nodes of the cortical grasping network send convergent projections to specific, overlapping territories of the putamen[16,17]. This nucleus serves as the main entry point of the basal ganglia, a system of deep brain structures of considerable clinical interest due to their crucial role in selecting and

[1]Department of Medicine and Surgery, Neuroscience Unit, University of Parma, Parma, Italy. [2]Present address: Center for Translational Neurophysiology of Speech and Communication, Italian Institute of Technology, Ferrara, Italy. [3]These authors jointly supervised this work: Monica Maranesi, Luca Bonini ✉e-mail: luca.bonini@unipr.it

controlling goal-directed and habitual actions[18–20]. The extensive cortico-striatal projections from the main nodes of the cortical grasping network to the putamen suggest that this nucleus may function as a key coordination hub for large-scale, functionally specialized networks dedicated to the selection and control of manipulative actions. Surprisingly, the possible contribution of putaminal neurons to the selection and organization of grasping actions remains unknown. Even less explored is their activity during the observation of other's action, which is known to induce a balanced facilitation and suppression of corticospinal neuron activity[21], as well as of neurons in the parietal[9] and premotor[11,22] cortical regions. These modulations may result from cortico-striatal loops hypothesized in previous studies[8,23], potentially enabling the selection or suppression of one's own motor actions while observing those of others—especially when social coordination in a shared space is required.

To address these issues, we designed a Mutual Action Task (MAT) in which a monkey and an experimenter, facing each other on opposite sides of a table, took turns grasping or observing their partner grasp the same multi-affordance object using either a precision grip (PG) or a whole-hand prehension (WH), based on a previously presented visual instruction. We recorded neuronal activity using multiple linear 32-channel probes from the sector of the putamen that receives anatomically-verified projections from cortical territories involved in forelimb motor control. The cortico-striatal synaptic input, indexed by

local field potentials (LFPs) modulations[24], faithfully reflected known cortical dynamics during sensory instructions and motor planning, while single-neuron activity selectively encoded the monkey's own action, the experimenter's action, or both. The grip type was clearly encoded by nearly a quarter of the neurons during the monkey's own trials, but not during the partner's trials. Control conditions revealed that viewing the partner's action was not necessary for most recorded neurons, as they still discharged even when the partner acted in the dark. Notably, a sizeable fraction of these neurons exhibited reduced or abolished discharge when the observed action occurred behind a transparent plastic barrier, supporting a pragmatic role for the monkey putamen in social coordination.

## Results

### Histological and connectional characterization of the recorded region

We used silicon and stainless-steel 32-channel linear electrode arrays ("Methods") in Mk1 and Mk2, respectively, employing a chronic (Supplementary Fig. 1a) or semi-chronic (Supplementary Fig. 1b) vertical approach. We targeted a region of the putamen corresponding to the stereotactic coordinates of the previously identified territory receiving projections from multiple areas of the cortical grasping network[17], as verified by post-mortem 3D reconstruction of the recorded regions (Fig. 1a). The recordings spanned the entire dorso-ventral extension of

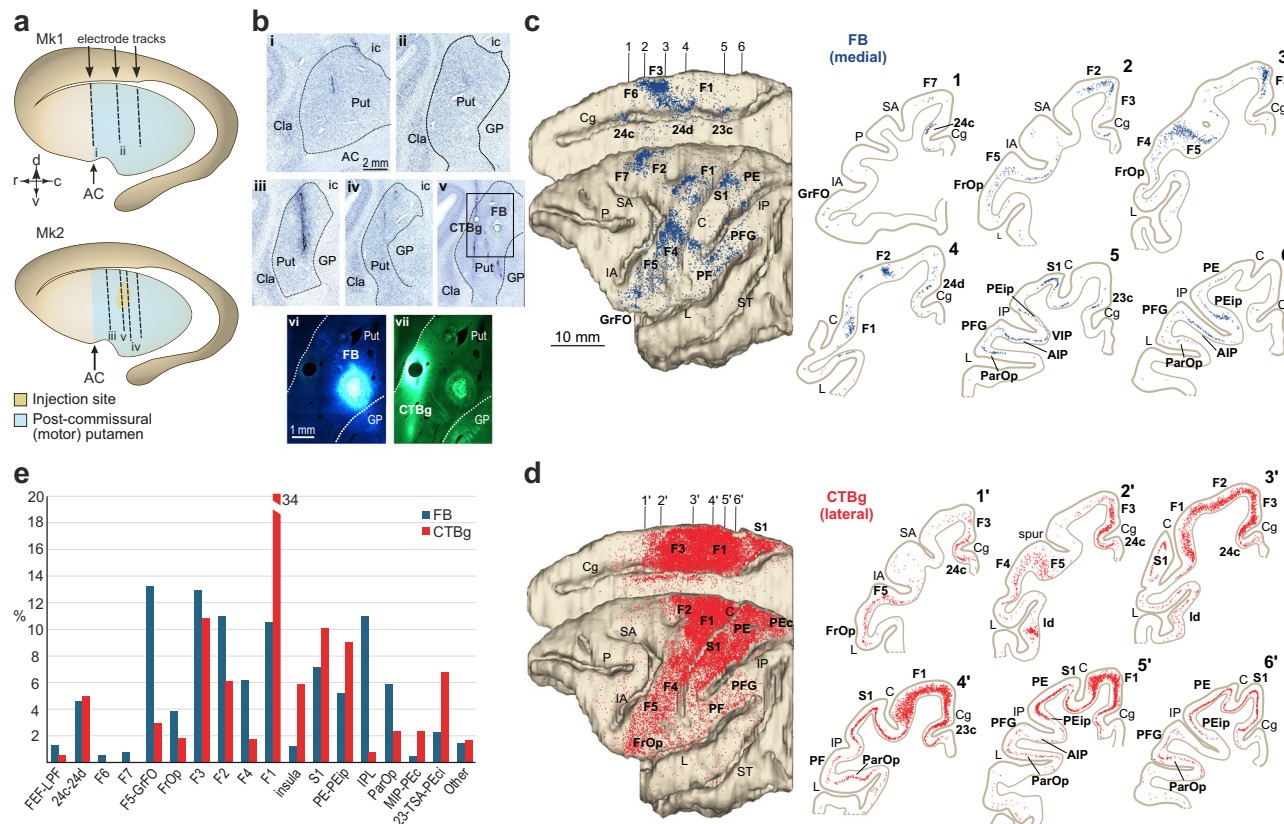

**Fig. 1 | Anatomical characterization of the recorded region. a** Reconstruction of the trajectory of implanted probes for each monkey. AC, Anterior commissure. **b**, Photomicrographs of Nissl-stained sections showing electrode tracks in the putamen of Mk1 (i, ii) and Mk2 (iii, iv), and injection tracks (v) and sites (F, Fast Blue, FB; and vii, cholera toxin subunit B conjugate with Alexa Fluor™ 488, CTBg) in Mk2. Cla, claustrum; GB, globus pallidus; ic, internal capsule; Put, putamen. Scale bar in i applies also to ii-v; scale bar in vi applies also to vii. Distribution of retrogradely labelled neurons following FB (**c**) and CTBg (**d**) injections in Mk2. Left: labelled neurons mapped on dorsolateral and medial views of a 3D reconstruction of the injected hemisphere. Right: coronal sections corresponding to numbered positions

arranged from rostral to caudal; each dot represents one labelled neuron. AIP anterior intraparietal area, C central sulcus, Cg cingulate sulcus, FrOp frontal operculum, GrFO granular frontal area, IA inferior arcuate sulcus, IP intraparietal sulcus, L lateral sulcus, P principal sulcus, ParOP parietal operculum, PEc caudal portion of area PE, PEip intraparietal portion of area PE, SA superior arcuate sulcus, ST superior temporal sulcus. **e** Percent distribution of FB- and CTBg-labelled neurons across cortical areas. FEF frontal eye field, IPL inferior parietal lobule, LPF lateral prefrontal cortex, MIP medial intraparietal area, TSA transitional sensory area. Histograms are shown only for areas where the proportion of labelled neurons exceeded 0.5% of the total.

the putaminal forelimb representation[25], confirmed by the reconstruction of electrode tracks in Nissl-stained coronal sections of the two monkeys' brains (Fig. 1b). In Mk2, we also conducted an anatomical tracing study at the end of the neurophysiological experiments to directly characterize the connectivity pattern of the core region (yellow in Fig. 1a) where neurons were previously recorded.

The results confirmed that the medial aspect of the putamen (Fig. 1c) receives projections specifically from the forelimb motor regions of the cingulate cortex[26], the mesial and dorsal premotor cortices[6], the ventral premotor cortex[10,27,28], the primary motor cortex[29], as well as from regions in the ventral and dorsal bank of the intraparietal sulcus and the inferior and superior parietal lobules[30,31]. The most lateral part of the putamen exhibits a similar connectivity pattern (Fig. 1d), with strong input from the primary motor and dorsal premotor cortices (Fig. 1e), supporting a prominent role of the putaminal investigated regions in forelimb motor control.

## Monkeys exploit the contextual instructions to perform the Mutual Action Task with an experimenter

We collected neuronal and behavioural data in 32 sessions ($n = 18$ in Mk1, $n = 14$ in Mk2) during the MAT (Fig. 2a). At the beginning of each trial, both partners gently pressed a button with their right hand (contralateral to the monkey's recorded hemisphere) and, in complete darkness, received an auditory cue instructing who was required to act

in the ongoing trial (*Agent cue*, Fig. 2a) and, next, a visual cue specifying which grip type (precision or whole-hand grip, *Grip cue*, Fig. 2a) to use for grasping a multi-affordance object (Fig. 2b). Afterward, the light turned on (*Object vision*, Fig. 2a) and when the cue sound ceased (Go signal) the previously instructed agent (i.e. monkey or experimenter) had to reach and grasp the object with the appropriate grip type and lift it (Action), while the partner remained still. In half of the trials, the light switched off at the onset of the reaching movement (*Action vision*, Fig. 2a), requiring the agent to perform the action in complete darkness. After each correctly executed trial, the monkey was automatically rewarded with a drop of juice (Fig. 2a). This task allowed us to assess the effect of three main factors—the agent identity (monkey and experimenter), the grip type (precision and whole-hand grip), and action visibility (full light and in the dark)—which, combined, resulted in 8 main conditions. We collected at least 15 correctly performed trials for each condition per session.

Behavioural evidence demonstrates an overall robust acquisition of the task rules by both animals (Fig. 2c), with an average percentage of correctly performed trials higher than 80% (83% for Mk1, 87% for Mk2). Specifically, monkeys acted and refrained from acting coherently with the auditory cue (Mk1, $\chi^2 = 2179$, $p = 10^{-475}$; Mk2, $\chi^2 = 2132$, $p = 1.7 \times 10^{-465}$) and correctly performed the previously instructed type of grip (Mk1, $\chi^2 = 1437$, $p = 1.22 \times 10^{-314}$; Mk2, $\chi^2 = 1102$, $p = 1.14 \times 10^{-241}$). Furthermore, the merely executive errors in task performance were

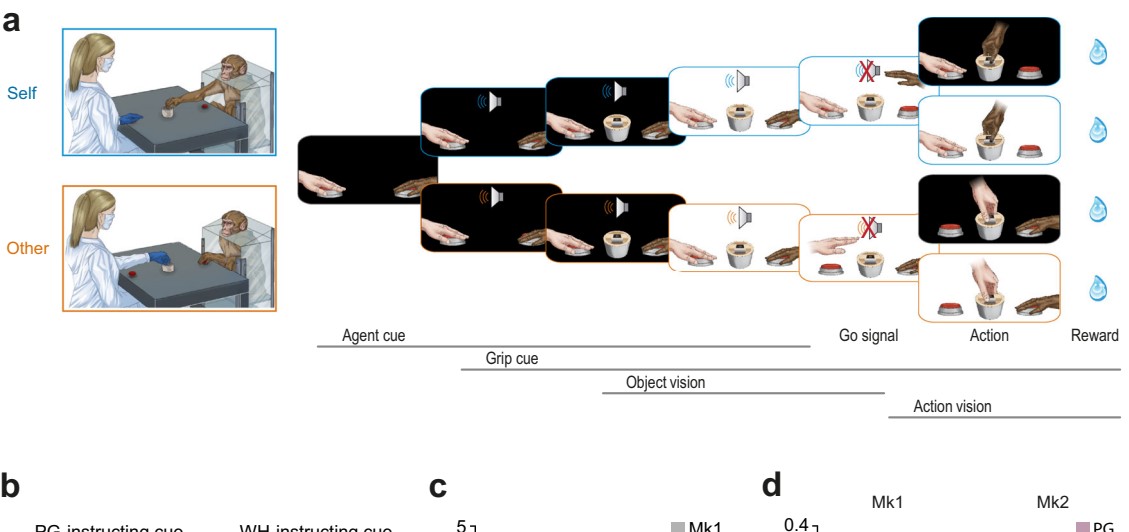

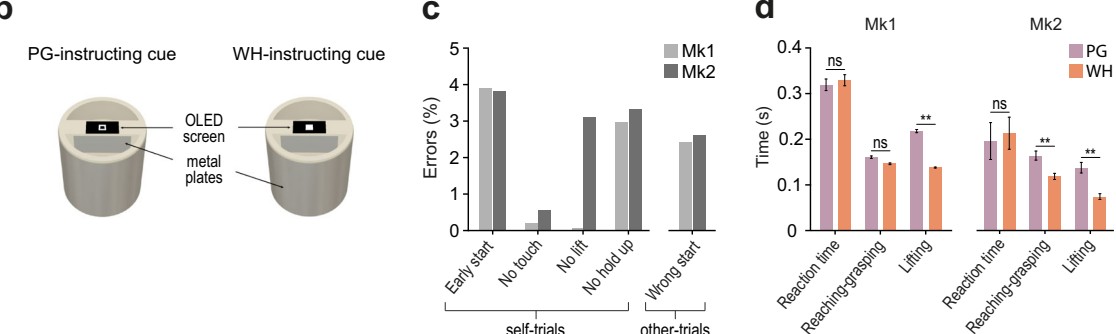

**Fig. 2 | Experimental setup and behavioral performance. a** Experimental setting of the MAT and temporal sequence of events for self and other trials. Each trial began with the agents keeping their hand still in the starting position in complete darkness (1 s). One of two auditory cues (300 or 1200 Hz sine wave), previously associated with each agent, was presented to indicate who had to perform the action (human or monkey, agent cue). After 770 ms, the type of grip to be used was indicated by a symbolic cue consisting of an empty (PG) or filled (WH) square that appeared on an OLED micro-screen installed on top of the object (grip cue). Finally, the environmental light was turned on (object vision) and, after 730 ms, the sound ceased (Go signal) and the instructed agent had to reach, grasp with the specified

grip, and then lift the object for at least 500 ms to obtain the reward. The same actions could be performed either in full light or in complete darkness (action vision). All trial types were fully randomized. **b** Schematic drawing of the multi-affordance object (see Methods) to be reached and grasped by each partner, either with a precision grip (PG; left panel) or a whole-hand prehension (WH; right panel), depending on the visual cue displayed by the OLED screen on top of the object. **c** Classification of error types (*Methods*) in both animals across all self- and other-trials of all the recording sessions. **d** Average (±1 SEM) reaction time and execution time for the different phases of the MAT in Mk1 ($n = 18$ sessions) and Mk2 ($n = 13$ sessions). Paired-sample t-tests ** $p < 0.001$.

very rare, and the monkeys made very few errors during the other's trials. These findings demonstrate a similar and high level of accuracy and compliance with the task instructions in both animals, during both the execution and observation tasks.

The analyses of reaction times, reaching-grasping times, and lifting times revealed a substantial similarity in the execution of the MAT between the two animals (Fig. 2d). For both monkeys, the reaction time did not differ between trials instructed with the PG or WH cue. In contrast, in PG trials, which require a higher degree of motor control relative to WH trials, the reaching time was significantly longer in Mk2 ($t = 8.47$, $p = 2.09 \times 10^{-6}$) and the lifting time was longer in both animals (Mk1: $t = 11.47$, $p = 2 \times 10^{-9}$; Mk2: $t = 9.11$, $9.66 \times 10^{-7}$).

### Local Field Potentials (LFPs), but not single neurons, encode contextual instructions

Since most of the input to the putamen nucleus derives from the cortex, and LFP signals are largely driven by synaptic modulations[24], we first assessed the changes in power within each the three main LFP frequency bands (low, 2–8 Hz; medium, 13–28 Hz; high, 60–100 Hz) during self and other trials of the MAT (Fig. 3a). The dynamics of LFP modulation observed during the MAT closely resemble those previously reported in motor cortical regions using various visuomotor paradigms[32–34].

During self-trials, a sizeable and progressively increasing proportion of recorded channels exhibited significant power suppression relative to baseline within the low and medium frequency bands as the task advanced from agent cue to grip cue, and finally to the visual presentation of the object; at that stage, virtually all channels showed marked suppression in both low and medium frequency bands (Fig. 3b). In contrast, power in the high frequency band significantly increased at the Go signal, with nearly all channels exhibiting modulation in this band during both the reaching and grasping phases of monkey's own action. Consistently, a subset of the modulated channels also displayed grip selectivity within the low and medium bands prior to movement onset, and in the low and high frequency bands during reaching and grasping (Fig. 3c).

Very similar modulations across the different frequency bands were observed during other-trials (Fig. 3d). However, no increase in low-band power was evident during the partner's action, and the enhancement in the high frequency band appeared earlier – starting at the Go signal and continuing into the reaching phase. Moreover, virtually no recording site showed grip selectivity during other-trials (Fig. 3e). We further investigated the relationship between LFP modulations during self and other trials by computing the correlation coefficients for each site and frequency (see *Methods*), where maximal self-other similarity is indicated by values approaching 1 and maximal

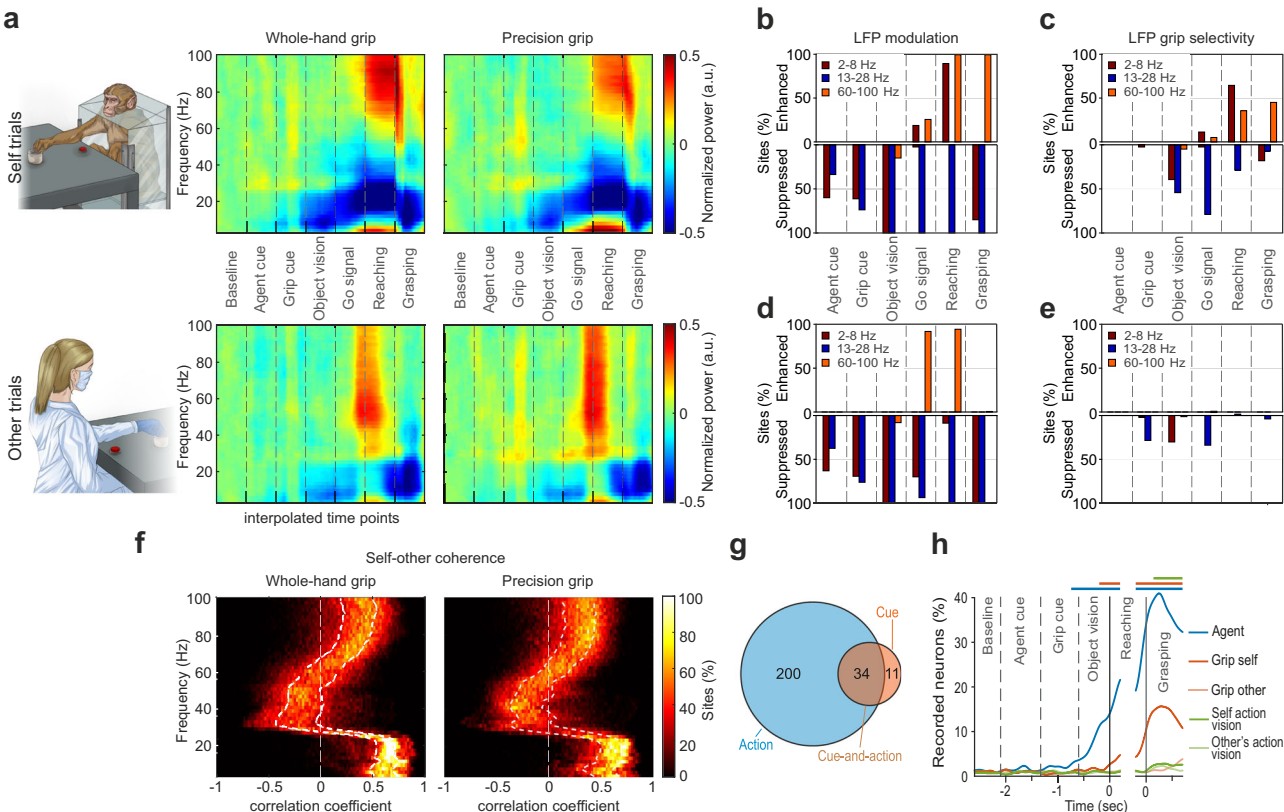

**Fig. 3 | Local Field Potentials (LFPs) and single-cell modulation during self and other trials. a** Time-frequency plots of LFP power modulation during self and other trials of the MAT, with WH and PG, averaged across all sites and sessions recorded with probe #1 from Mk1. LFP modulation was generally consistent across recording sites (Supplementary Fig. 2a, b) and sessions (Supplementary Fig. 2c). **b** Percentage of the total recorded sites ($n = 160$, 32 sites × 5 probes) during self-trials showing significantly enhanced (top) or suppressed (bottom) LFP power relative to baseline within the low-, medium-, or high-frequency bands (Kruskal-Wallis test followed by one-tailed Wilcoxon test). **c** Percentage of the total recorded sites showing significantly enhanced (upper panel) or suppressed (lower panel) LFP power in PG relative to WH trials within the low-, medium-, or high-frequency bands. Statistical analysis as in (**b**). **d** Same as b, for other-trials. **e** Same as (**c**) for other-trials.

**f** Percentage of recorded sites showing significant positive (to the right of the white dashed line) or negative (to the left of the white dashed line) correlations between self- and other's action performed with whole-hand (left) or precision grip (right), as a function of LFP frequency bands. The two white dotted lines represent the 25th and the 75th percentiles, respectively. **g** Number of recorded neurons encoding the action, the cue, or both. **h** Percentage of recorded neurons showing selectivity for agent identity, grip type (during self and other trials), and vision of one's own or the other's hand during task unfolding. Continuous colored bars above the plot indicate the period in which the fraction of selective neurons was significantly higher than during baseline (two-tailed sliding t-test, $P < 0.01$). Exact p-values for each population-level comparison are reported in the Source Data file.

dissimilarity by values approaching −1. Most sites exhibited robust and positive correlations in the lower frequency bands (<30 Hz) during both PG- and WH- grip trials (Fig. 3f).

Interestingly, while LFP modulations in the low and medium frequency bands encoded contextual task cues, this was not the case for high frequency bands, which are thought to reflect neuronal spiking activity[24]. Indeed, single-neuron analyses (see *Methods*) revealed that among 560 isolated neurons (415 from Mk1 and 145 from Mk2) 245 were task-related, and very few of them exhibited purely cue-related activity ($n = 11$). Some were modulated during the agent and/or grip cues as well as during the subsequent action execution phase ($n = 34$), whereas the vast majority ($n = 200$) displayed purely action-related responses (Fig. 3g). By looking at the tuning of the recorded neurons in a time-resolved manner (200-ms sliding window in steps of 20 ms), no significant coding for contextual task cues was detected before the

object became visible. Notably, the highest fraction of neurons exhibiting selectivity for agent identity, grip type, and action vision was found during the object grasping phase (Fig. 3h).

Taken together, these findings suggest that cortico-striatal synaptic input may convey early contextual information but influences putaminal spiking activity primarily during self and other's action. We therefore focused subsequent analyses on self- and other-related neuronal responses during the movement epochs of the MAT.

## Putamen neurons encoding self and other's action

Action-related neurons were classified based on their response to actions performed by the two agents. Self-type neurons (ST, $n = 153$, 65%) were modulated exclusively during the monkey's own action (Example neuron 1, Fig. 4a). Other-type neurons (OT, $n = 30$, 13%) responded only to the experimenter's action (Example neuron 2,

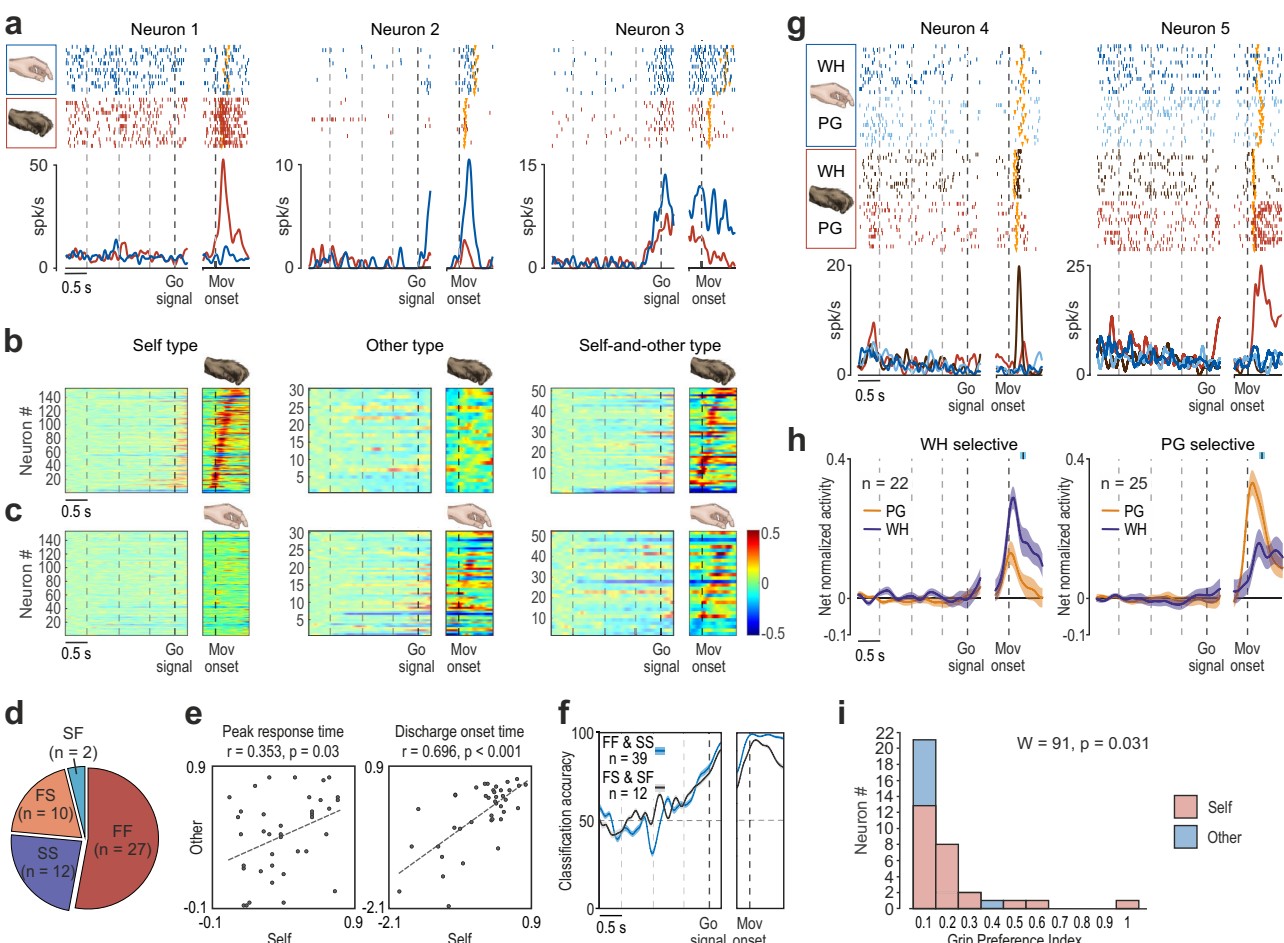

**Fig. 4 | Neuronal properties of self-type, other-type, and self-and-other-type putaminal neurons. a** Examples of self-type (Neuron 1), other-type (Neuron 2), and self-and-other-type (Neuron 3) cells recorded during monkey's own (red) and experimenter's (blue) actions. Rasters and peri-stimulus polyline plots are aligned on the Go signal (left) and (after the gap) on the Movement onset (right). Dashed lines on the left part of the plot indicate the different cues provided during the MAT. Orange triangular markers indicate the moment when the (experimenter's or monkey's) hand touched the target. **b** Activity heat maps of self-type, other-type, and self-and-other-type neurons during monkey trials. Alignments as in (**a**). Neurons are ordered (from bottom to top) based on the timing of their peak of activity during monkey's own action execution, except for other-type neurons, which have been ordered as described in (**c**). **c** Heat maps of self-type, other-type, and self-and-other-type neurons during experimenter's trials. Other-type neurons have been ordered (from bottom to top) based on the timing of their peak of activity during experimenter's action, while the other neuronal population have been ordered as described in (**b**).

**d** Number of neurons with different combinations of facilitated (F) and suppressed (S) responses during monkey's and experimenter's trials, respectively. **e** Scatterplots illustrating the relationship between the peak of activity (left) and discharge onset (right) timing of FF and SS SOT neurons during self and other trials, calculated relative to Movement onset (Methods). Pearson's correlation test, two-sided. **f** Classification accuracy of agent identity obtained with a Bayesian classifier trained and tested with the activity of two different sets of SOT neurons: neurons coherently (FF and SS, blue) and oppositely (FS and SF, black) modulated during self and other trials. Data are presented as mean values ± SEM. **g** Example neurons with grip selectivity during self-trials. Conventions as in (**a**). **h** Population activity time course for subsets of neurons selectively encoding WH or PG during self-trials. The black marker with light-blue shading above each plot indicates the average ± 1 standard deviation of object contact during action execution relative to Movement onset. **i** Frequency distribution of grip Preference Index values for FF SOT neurons response during self- and other's actions.

Fig. 4a). Self-and-other-type neurons (SOT, $n = 51$, 22%) responded during the action of both agents (Example neuron 3, Fig. 4a). Most action-related neurons exhibited facilitated discharge during self-trials (Fig. 4b), whereas those responding during other-trials showed a more balanced distribution of facilitation and suppression (Fig. 4c). Facilitated neurons displayed a sharp increase in firing rate tightly aligned with movement onset, particularly during self-trials. In contrast, suppressed neurons exhibited an earlier and more gradual reduction of their activity, especially during other trials (Supplementary Fig. 3). These findings suggest that while self-type neuronal responses are likely driven by direct cortico-striatal projections[35], other-type responses may be more strongly shaped by local inhibitory mechanisms. Interestingly, all but one of the other-related neurons (OT and SOT combined) were recorded simultaneously with at least one self-type neuron (Supplementary Fig. 4). Thus, despite the lack of electromyographic recordings, we could verify the absence of modulation among ST neurons during other trials, which makes it unlikely that other-related responses were due to covert muscle activation, consistent with previous direct evidence from monkey action observation studies[11,36–38].

Examining the relationship between SOT neuron response during self and other trials revealed that the majority (73%) exhibited a coherent facilitation (FF, $n = 24$) or suppression (SS, $n = 13$) in both contexts, while the remaining neurons displayed opposite modulations (FS and SF, Fig. 4d). Considering SOT neurons with coherent modulations during self and other trials, the timing of peak activity and discharge onset were not significantly different (peak, $t = -0.31$, $p = 0.76$; onset, $t = -0.37$, $p = 0.71$) and showed moderately positive and significant correlations (Fig. 4e). These findings emphasize the similarity in SOT neuron dynamics across self- and other-trials. Nevertheless, using a Bayesian classifier trained to distinguish self from other trials ("Methods"), we found that SOT neurons exhibiting coherent facilitation or suppression (FF and SS) carried as much information about agent identity as SOT neurons modulated in an opposite manner (FS and SF) for the two agents (Fig. 4f). This indicates that even neurons with similar responses to both agents' actions retain information about *who* is acting, highlighting a strong and widespread agent-specific coding in primate putamen neurons.

Next, we investigated whether agent-specific coding of actions also reflects the way in which grasping is performed—namely, using either a PG or a WH grip—as previously demonstrated in all cortical areas projecting to the putamen[5,27,39]. Among action-related neurons active during the monkey's own actions ($n = 204$), approximately one quarter (23.5%) encoded grip type, with a similar fraction preferring PG ($n = 30$, 14.7%, example Neuron 4, Fig. 4g) and WH ($n = 24$, 11.8%, example Neuron 5, Fig. 4g). As the vast majority of grip-selective neurons exhibited facilitated discharge ($n = 47$, 87%), we further examined the population dynamics of these neurons (Fig. 4h), confirming the absence of grip-type coding prior to movement onset and a sharp tuning to grip type from reach onset through grasp execution. Importantly, grip-selective neurons did not significantly vary their tuning depending on whether the action was performed in the light or in the dark (Supplementary Fig. 5), supporting the inherently motor nature of their discharge. Among action-related neurons active during the experimenter's actions ($n = 81$), only a few ($n = 8$, 9.9%) encoded the grip type. Two of these were SOT neurons, but their grip selectivity was different during self and other trials. Thus, although putamen neurons display significant tuning for grip type during monkey's own actions, they do not generalize this coding feature to others' actions (Fig. 4i).

## No need to see, but to share the space for action

The visual input from the monkey's own hand during action execution exerts a facilitatory effect on the discharge of parietal[40] and premotor[37] neurons, particularly when comparing those responsive to the action

of others with purely motor neurons. This finding suggests that cortical SOT neurons process visuomotor information regarding both self and others. However, whether this property also applies to putaminal neurons—which are direct targets of projections from the core areas of the cortical grasping network—remains to be determined.

To address this issue, we compared neuronal responses during MAT trials performed by the monkey in the dark and in full light (Fig. 5a), allowing us to manipulate the visibility of the monkey's own action. Among 204 neurons responding during monkey's own action, the vast majority showed no significant difference in discharge intensity between the light versus dark condition (Fig. 5b and example neuron 6, Fig. 5c), regardless of whether they were responsive (SOT) or non-responsive (ST) to other's action ($c^2 = 0.6$, $p = 0.44$), in striking contrast to previous findings reported in the parietal and premotor areas[37,40]. Even at the population level the response of neurons facilitated ($n = 155$, 76%) and suppressed ($n = 49$, 24%) during monkey's own action showed no significant difference between grasping in the light and in the dark (Fig. 5d), indicating that putaminal neuron activity is essentially unaffected by the visual feedback from monkey's own acting hand.

Next, we asked whether visual information was actually necessary for putaminal neurons to encode the partner's action. In fact, we can easily be aware of another's action even if it occurs in the dark or behind an occluder, provided that prior visual or auditory cues are available, and neurons responsive to others' observed actions can fire with a notable degree of independence from actual action visibility[36,38,41,42], suggesting remarkable generative properties within the cortical action observation network[23]. To address this issue in the putamen, we compared neuronal responses during MAT trials performed by the experimenter in the dark and in full light (Fig. 5e). Among 81 neurons responsive during the partner's action in the light (SOT, $n = 51$; OT $n = 30$), the overwhelming majority ($n = 76$, 94%) showed no difference in their discharge when the experimenter's action occurred in the dark (Fig. 5f and example neuron 7, Fig. 5g). Even at the population level, the response of both facilitated ($n = 44$, 54%) and suppressed ($n = 37$, 46%) neurons during partner's action showed no significant difference between light and dark conditions (Fig. 5h), indicating that the encoding of others' action by neurons of the motor putamen can occur entirely independently of visual information.

Finally, given their marked independence from visual input, we aimed to further scrutinize the relevance of putaminal neuron activity during the partner's action. Based on substantial evidence from neurophysiological studies of the areas belonging to the cortical grasping network[7,9,10,15], we hypothesized that other-related neural responses in the striatal node of the cortico-basal ganglia loop play a pragmatic role, being modulated by the possibility of interacting with the target object. To directly test this hypothesis, we employed an additional experimental condition in which the experimenter's action was performed in full light behind a transparent plastic barrier interposed between the monkey's hand and the target (Fig. 6a). Note that the barrier did not alter neither the task contingencies nor the available visual information with respect to the social condition, but introduced a physical separation between the monkey and the experimenter's target, rendering it unreachable for the monkey. Out of 70 putaminal neurons tested in these conditions (42 SOT and 28 OT, Fig. 6b), 22 (31.5%) discharged similarly with and without the barrier, 5 increased their firing with the barrier (7.5%), while the response of the remaining (61%) was either reduced ($n = 8$, 11%), or abolished ($n = 35$; 50%; example neuron 8, Fig. 6c) during the barrier with respect to the social condition. Population responses showed a similar effect of the barrier among facilitated and suppressed neurons (Fig. 6d), indicating that the mere vision of the partner's action is not only unnecessary, but even not sufficient to trigger putaminal neuron activity. Rather, what appears critical is the possibility for the monkey to interact with the

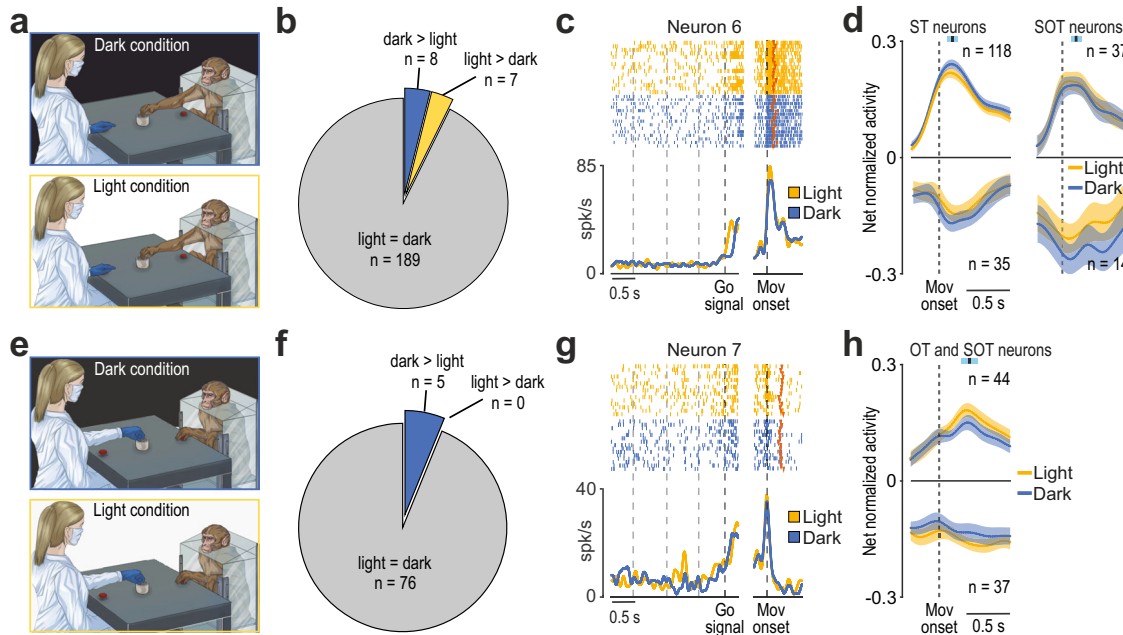

**Fig. 5 | Relevance of visual information for putamen neuron activity.**
**a** Schematic drawing of the MAT in dark and light conditions during the monkey's trials. **b** Fraction of neurons influenced by the visual feedback of the monkey's hand during grasping execution. **c** Example of the motor response of a putamen neuron discharging similarly during grasping in both light and dark conditions. Conventions as in Fig. 4a. **d** Population activity of ST and SOT neurons during monkey trials in light and dark conditions. Vertical dashed lines (alignment point) indicate movement onset. Colored lines represent net normalized activity ± 1 standard error. The black marker with light-blue shading above each plot indicates the average ± 1 standard deviation of object contact during action execution relative to movement onset. **e** Schematic drawing of the MAT in dark and light conditions during the experimenter's trials. **f** Pie chart illustrating the fraction of neurons influenced by the vision of experimenter's hand. **g** Example of the response of a putamen neuron discharging similarly during experimenter trials in both light and dark conditions. Conventions as in (**c**). **h** Population activity of OT and SOT neurons during the experimenter's trials in light and dark conditions. Conventions as in (**d**).

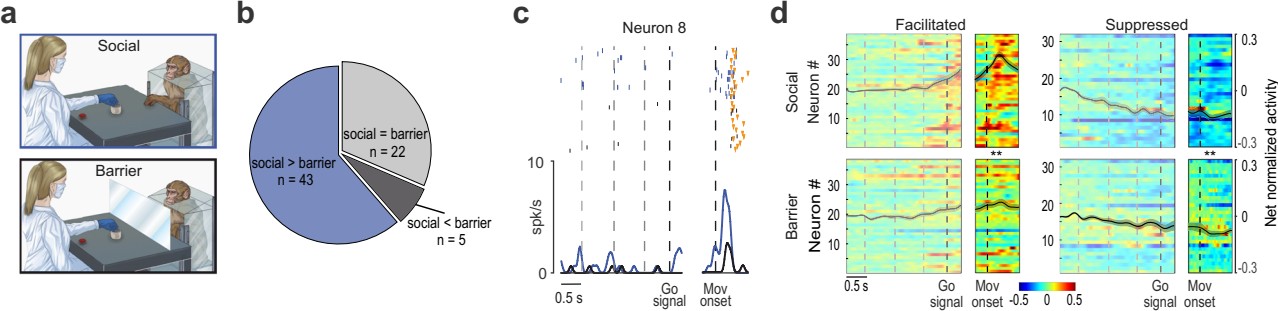

**Fig. 6 | Partner's action behind a barrier. a** Schematic drawing of the MAT in the Social and Barrier conditions during experimenter's trials. **b** Pie chart illustrating the fraction of OT and SOT neurons influenced by the presence of the barrier. **c** Response of a representative putaminal neuron discharging only during the experimenter's grasping in the Social condition (blue). Other conventions as in Fig. 4a. **d** Heat maps of facilitated and suppressed neurons during experimenter's trials in the Social (top) and Barrier (bottom) conditions. Conventions as in Fig. 4b. Neurons have been ordered (from bottom to top) based on the timing of their peak of facilitated or suppressed activity during experimenter's action in the Social condition. Statistical difference between Social and Barrier condition has been assessed by mean of a repeated-measures ANOVA followed by Tukey's post hoc tests. ** $p < 0.05$.

target of the partner's action, thereby highlighting the pragmatic nature of other-related putaminal neuron response.

## Discussion

We leveraged a Mutual Action Task involving a monkey and a human partner to uncover a possible role of the primate putamen nucleus in representing manual actions of self and others. We found that local field potentials in the motor region of the putamen nucleus exhibit modulations during the sensory cues indicating who should act and what action to perform, similar to those previously reported in the spiking activity of areas of the cortical grasping network projecting to it[4,10,43,44]. In contrast, spiking activity in this portion of the putamen did not reflect prior sensory cues, but selectively encoded self- and other's

actions and the grip type employed by the monkey to grasp the target. Notably, the encoding of both self and other's actions was unaffected by the absence of visual input, when the action was performed in the dark; yet others' action coding was disrupted by interposing a physical transparent barrier between the monkey and the partner's target. This manipulation did not alter the available visual information but specifically eliminated the monkey's opportunity to physically interact with the object, thereby supporting a pragmatic coding[7,45] of others' actions by primate putaminal neurons.

Classical models of primate cortico-basal ganglia circuits have primarily focused on forelimb reaching[46] and reward-based[17] action selection, leaving the putamen's role in manual action underexplored. We provide direct evidence that putaminal neuron discharge encodes

monkeys' reaching-grasping actions, sometimes with remarkable grip-type selectivity even in the absence of vision, thereby confirming its motor nature. In addition, our anatomical data show that the sector of the putamen where neurons were recorded receives direct projections from the frontal and parietal regions of the cortical grasping network[5,27,39], in line with previous findings[17]. Altogether, our anatomo-functional results provide direct support for extending the network of brain areas involved in the planning and control of manual actions to include the putamen nucleus.

Coherently with the anatomical evidence, low-frequency local field potentials (LFPs), which likely reflect long-range cortico-striatal inputs, encode the sensory cues provided to both agents prior to movement onset, likewise neurons in the cortical areas projecting to them[4,47]. Furthermore, these bands exhibit the highest inter-agent coherence during object grasping, highlighting the possible role of cortico-striatal input in the temporal parsing of self- and other-generated motor events. In striking contrast, high-frequency LFPs as well as spiking activity primarily encode agent identity and grip parameters during action execution but not during the sensory cue period[48], suggesting that while parieto-frontal circuits specify a variety of action possibilities, activity in the motor putamen plays a specific role in selecting the current option to be turned into action. These findings support a role of the basal ganglia in the "affordance competition"[1], extending its relevance beyond object-directed actions to encompass the social coordination required during interaction with others[2,14].

Indeed, so far, the possible involvement of the putamen nucleus in representing the action of others has been limited to a few neuroimaging[49] and indirect neurophysiological[50] studies in humans, leaving single-neuron mechanisms unexplored. Our data not only demonstrate the existence of putaminal neurons responding during another's action, but also emphasize a remarkable difference in grip-type selectivity between self- and other-trials. Indeed, hand-grip tuning emerged during the execution of the monkey's own grasping actions, whereas it was considerably reduced during the partner's trials and, when present, did not match the motor selectivity. The lack of congruence in single-neuron grip tuning between self and other trials is consistent with previous findings in the premotor[51] and parietal[9,52] cortices. However, the proportion of putaminal neurons with visual (10%) and motor (25%) selectivity for the grip type is considerably lower than that reported for all cortical areas for which data are available. Together with our anatomical connectivity data, these findings suggest that within the cortico-striatal system, cortico-cortical sensorimotor circuits specify the detailed features of potential motor plans (e.g., grip type) based on object- or agent-specific affordances, whereas the basal ganglia modulate the decision of whether or not to respond to another agent's action – and not necessarily with a matched action[14,53].

Our control experiments clarified the potential role of putaminal neurons encoding others' actions. Nearly all recorded neurons remained responsive to the experimenter's action even when the monkey could not see it due to the light being switched off. This effect was markedly stronger than previously reported at the cortical level, where only about half of the recorded neurons responded in the absence of direct visual information[41]. Moreover, visual information alone was neither necessary nor sufficient to trigger putaminal neuron response. In fact, when the experimenter's action was fully visible but occurred behind a transparent barrier that precluded any potential interaction between the monkey and the target, the neuronal response was largely abolished, supporting the view of a pragmatic, rather than visual[9] or metric[7] coding of other's actions by putaminal neurons.

Possible confounds and alternative interpretations of our findings must also be considered. One potential confound is that the responses of other-related neurons might, in fact, reflect subtle or covert muscle activation. Although electromyographic activity was not recorded in

the present study, several previous reports have consistently demonstrated that when monkeys observe another individual performing predictable actions, no muscle activity is detectable[11,21,36,37]. Consistent with this evidence, our simultaneous recordings (Supplementary Fig. 4) revealed that ST neurons− which are active during the monkey's own actions, when muscle activity is high−remain silent during other's actions, precisely when simultaneously recorded other-related neurons discharge. These observations strongly argue against the interpretation that the activity of other-related neurons is accounted for by covert motor activation. A second potential confound derives from our choice to perform head-free recordings to increase the ecological validity of the results[54], which prevented us from monitoring eye movement to ensure that monkeys were attending to the partner's action. Although monkeys were required to attend to the workspace shared with the experimenter to correctly perform the task, it remains possible that visual attention or stimulation differed between self and other trials, thereby influencing neuronal discharge. However, previous studies have shown that direct gaze toward the partner's action is not required to elicit other-related neuronal responses in cortical areas that provide input to the putamen[51,55]. Most importantly, putamen neurons discharging during self and other trials were largely independent of visual information, making it unlikely that gaze or attention played a major role in shaping their activity. Closely related to this latter point, an alternative interpretation of SOT neuron activity is that their responses reflect a timing signal linked to the predictable delivery of reward rather than a motor signal. This interpretation is consistent with the similarity in SOT neuron response dynamics observed across self and other trials. However, the absence of responses in the barrier condition, where the temporal relationship with reward remained unaltered, stands in sharp contrast with this interpretation and instead supports our pragmatic hypothesis. Altogether, these findings highlight the contribution of other-related processes to action selection during social contexts and suggest extending the recently proposed social affordance hypothesis[2,53] to include the putamen and the cortico-striatal circuits as essential component of a cortico-subcortical social interaction network.

Considering the clinical relevance of the cortico-basal ganglia circuits, the present findings open new avenues for translational research. Emerging evidence indicates that dopamine and its primary targets within the basal ganglia circuits play a crucial role in regulating social versus non-social motor behaviors. Studies in Parkinson's patients comparing states on- and off-dopamine replacement therapy suggest that dopaminergic input to the basal ganglia enhances sensitivity to social cues and modulates kinematic patterns when planning and executing actions with social[56], cooperative[57], or communicative[58] intent, as opposed to individual or competitive actions. Another study showed that Parkinson's patients synchronized their grasping movements more effectively with a virtual partner during the drug-on condition compared to drug-off, supporting a key role of dopaminergic innervation of the cortico-striatal system in promoting flexible social behavior[59]. To date, the potential impact of dopaminergic modulation on self-, other-, and self-and-other-type striatal neurons remains unknown. Nevertheless, the present findings provide a starting point for future investigations aiming to develop a comprehensive neurochemical and anatomo-functional model linking cortico-cortical circuits for biological action processing with cortico-striatal loops involved in selecting motor responses to others during social interactions.

## Methods
### Animal models
This study involved two purpose-bred, socially housed adult male monkeys (*Macaca mulatta*, 9 and 12 Kg). Prior to recordings, the monkeys were habituated to sitting in a primate chair and interacting with the experimenters. Following this initial phase, they underwent

specific positive reinforcement training to perform the Mutual Action Task (MAT), described below, using the hand contralateral to the hemisphere to be recorded.

In preparation for neural data collection, the monkeys underwent a surgical procedure for the implantation of a recording chamber over the region of interest, leaving the skull intact. The recording probes were implanted during subsequent surgeries. All surgeries were performed in stereotaxic and aseptic conditions, under general anaesthesia induced by intramuscular injection of ketamine (5 mg/Kg) and medetomidine hydrochloride (0.05 mg/Kg) and maintained with 2% isoflurane vaporized in 100% oxygen. The monkey's vital parameters were continuously monitored with a multiparametric monitor. To ensure hydration, a constant intravenous infusion of saline solution was provided, and vitamin A gel was applied to prevent eye dryness during anaesthesia. Monkeys received analgesics, broad-spectrum antibiotics, and anti-inflammatory drugs during and after surgical procedures. Multielectrode linear probes were subsequently implanted in both monkeys during different surgeries.

All experimental procedures were conducted in agreement with the European (Directive 2010/63/EU) and Italian (D.lgs 26/2014) legislation for the Protection of Animals Used for Scientific Purposes, received the approval of the Veterinarian Animal Care and Use Committee of the University of Parma and of the Italian Ministry of Health.

## Behavioural paradigm

During the MAT, each monkey sat in its primate chair positioned at one end of a table, with the experimenter sat in front of it on the opposite side, taking on the role of a collaborative partner. Both subjects shared an operational space containing a multi-affordance object (Fig. 2b) that could be reached and manipulated by both partners. The object (5.5 cm-diameter, 6 cm-high cylinder, centrally topped by a 4.5 cm × 4.5 cm × 1.5 cm parallelepiped) positioned 16 cm away from each subject's initial hand position, afforded two distinct grip types: a precision grip (PG), involving thumb-index pinching on the central part of the object, and a whole hand prehension (WH), requiring wrapping the hand around its cylindrical body. A visual cue displayed on a 2 × 1 cm OLED screen embedded in the central part of the parallelepiped on top of the object indicated the required grip type: an empty square for PG and a filled square for WH. Metallic plates around each component of the object activated distinct capacitive circuits upon contact, thereby triggering a specific TTL signal for effective execution of PG and WH, recording and storing the detected contact times along with other behavioural and task events.

Each monkey was trained separately with the experimenter to interpret a sound as a Go or No-Go cue: a high-pitched tone (1200 Hz sine wave) instructed Mk1 to act and Mk2 to remain still; conversely, a low-pitched tone (a 300 Hz) instructed Mk2 to act and Mk1 to remain still. Each monkey's No-Go cue served as the Go cue for the experimenter, who performed the action while the monkey remained still.

Each trial began in complete darkness, with the monkey and the experimenter holding their right hand on a starting button. After 1 s (inter-trial interval), the cue sound was presented (*sound onset*), conveying an opposite Go or No-Go instruction to the partners. 770 ms after sound onset, the OLED screen displayed either an empty or a full square, instructing the agent to perform a PG or WH, respectively. After 730 ms, ambient light was turned on (*light onset*), making the object visible. Following an additional 570 ms, the cue sound ceased (*go signal*), indicating to the partner for whom the sound served as the Go cue to release the button, reach for, and grasp the object using the specified type of prehension within 1 s. The object had to be held for at least 500 ms in order for the monkey to receive the reward. In half the trials, the light it was switched off upon button release to eliminate visual feedback during monkey's or experimenter's movement (*dark condition*); in the other half, the light remained on until task completion (*light condition*). In addition to the basic *social condition* in which

the MAT could be performed by both partners, we also included a control condition in which a transparent plastic barrier was interposed between the monkey's hand and the target (*barrier condition*), thereby making it impossible for the monkey to interact with the target.

Whenever the monkey successfully completed all steps of a Go trial or remained motionless during a No-Go trial while the experimenter performed the actions, a fixed amount of liquid reward was automatically delivered; otherwise, the trial was aborted. A trial was considered correct only if both partners behaved appropriately. The task phases were automatically controlled and monitored using a LabView-based software, which also enabled trial termination in the event of behavioural error by either partner. Specifically, the monkey could commit the following types of errors during self trials: *early start*, releasing the button before sound offset; *no touch*, releasing the button without contacting the target object on time; *no lift*, touching the object but failing to lift it; *no hold up*, lifting the object but failing to keep it pulled up for at least 500 ms. During experimenter's trials, we evaluated possible *wrong start* errors, consisting in detaching the hand from the starting button during the experimenter's trials.

In each session, we collected 15 trials for each of the 8 experimental conditions (2 grip types × 2 agents × 2 visual feedback conditions), resulting in 120 correctly performed trials.

## Neural recording techniques

Each monkey was surgically implanted with a custom-made biocompatible plastic recording chamber (Mk1: 4.5 × 5 × 2.5 cm; Mk2: 2.8 × 3.3 × 2.5 cm), designed from 3D cranial reconstructions based on 7 T MRI data using 3D Slicer. Chambers featured parallel grooves (1 mm-wide, 2 mm-spacing) housing up to 8 (Mk1) or 4 (Mk2) Omnetics connector blocks interfacing multielectrode arrays and the wireless data logger headstages (Deuteron Technologies Ltd) for neural data collection. After full recovery from chamber implantation, different probe insertion strategies were employed in the two animals.

In Mk1, five 32-channel linear silicon probes (ATLAS Neuroengineering, IrOx contacts, 250 mm inter-site spacing, 24 mm length, 30 × 100 mm cross-section, 0.23–0.29 MW impedance) were chronically implanted through a small craniotomy performed within the chamber in two different surgeries. We used a dedicated insertion device[60] to position and release probes with a vertical approach at the desired location based on MRI-based reconstruction of stereotaxic coordinates. Probes featured a pointed tip to minimize tissue dimpling and were connected via flexible polyimide ribbon cables to the Omnetics connectors[61]. A collagen-based dural regeneration matrix (DuraGen® Plus) was placed around insertion sites, and liquid bone cement was then poured into the recording chamber, which rapidly solidified to secure the probes in place and prevent any contamination. Two months after the first surgery and the subsequent recording period, another surgery was performed to explant the first two probes and implant three new probes of the same type.

In Mk2, during the chamber placement surgery, bone cement was poured into the chamber (13-mm thickness) and allowed to solidify. Then, during two distinct surgeries spaced four months apart, four polyimide guide tubes (36 mm in length; outer diameter: 820 mm; inner diameter 760 mm) were implanted through the cement and the intact bone with a stereotaxically manipulated drill bit to ensure vertical trajectory to reach the desired depth and location. The polyimide tubes served as permanent guides, allowing for the precise, repeated insertion of individual linear multisite probes (32-channel Pt/Ir Plexon pig-tailed V-Probes, length: 45 mm from the first recording site; interelectrode spacing: 200 μm; recording site diameter: 15 μm). Each probe was sheathed in a stainless-steel ground tube and connected to an Omnetics interface via flexible cable. Probes were inserted weekly in the awake animal and semi-chronically fixed using Kwik-Cast™ silicone sealant.

Once the logger was connected to the electrode arrays, all components were enclosed in a protective chamber-mounted cap[54]. Neural signals were bandpass-filtered (2-7000 Hz), digitized at 32 kHz and stored locally on a 64 GB MicroSD card. The logger communicated with a PC via a USB-connected transceiver featuring four digital inputs and one digital output to synchronize neural and behavioral data.

### Recording of behavioural events and definition of the epochs of interest

Behavioural events were detected using distinct contact-sensitive devices that signalled 1) the detachment of each subject's hand from the starting button, 2) the hand-target contact (depending on the type of prehension), and 3) the object lifting. These events generated specific TTL signals, which were sent to a PC equipped with a dedicated LabView-based software that monitored the behavioural performance and controlled the generation of instructional cues and reward delivery. All input and output signals were synchronously recorded with the neural data for subsequent statistical analysis.

Neural signal analyses considered the following epochs defined based on task events: (1) *baseline*, ranging from 600 to 100 ms before the cue sound presentation, when the monkey was still with its hand on the starting position in the dark, prior to trial onset; (2) *agent cue* and (3) *grip cue*, both defined as the 500 ms following the sound and the grip cue presentation. The epoch starting with light onset was not included in the analysis because it made it possible to predict the Go/No-Go signal, hence likely included movement preparation. During the motor phase of the MAT, we identified the following additional epochs of interest: (1) *pre-movement*, corresponding to the 300 ms interval before the hand detached from the starting button; (2) *reaching-grasping*, corresponding to the interval ranging from the detachment of the hand from the starting button to target object contact; and (3) *lifting-holding*, ranging from object contact to 300 ms after this event. We ensured that we restricted the analysis only the period ending 300 ms after the contact with the object in order to encompass the initial lifting and holding phase while excluding any potential response associated with arm retraction or reward delivery[62,63].

### Local field potentials analysis

All signal analyses were performed offline using custom software written in MATLAB (MathWorks, Natick, MA, United States). LFP recordings were downsampled from 32 kHz to 1 kHz and band-pass filtered between 2–100 Hz using a second-order Butterworth filter, along with a 50 Hz notch filter (Q = 35). Time-frequency decomposition (Fig. 3a) was carried out using the multitaper method (Chronux toolbox http://chronux.org/; time-bandwidth product TW = 3, $K$ = 5 tapers), with 300-ms a moving window and a step size of 30 ms. The resulting spectral power was log-transformed by converting it to decibel (dB), calculated as ten times the base-10 logarithm of the spectral power magnitude at each frequency. Power values were segmented into the pre-defined epochs and interpolated to a uniform time width for trial alignment. For baseline normalization, the mean power from a 500-ms pre-agent cue baseline window was subtracted:

$$P_{norm}(f, t) = P_{dB}(f, t) - P_{dB}^{baseline}(f) \tag{1}$$

Then, for visualization and comparison across trials, sites and configurations, the values were scaled to the [−1, 1] range using the trial-wise maximum absolute value:

$$P_{scaled}(f, t) = \frac{P_{norm}(f, t)}{\max(|P_{norm}(f, t)|)} \tag{2}$$

To assess whether LFP power varied significantly across task epochs (Fig. 3b–e), a non-parametric one-way ANOVA (Kruskal–Wallis test) was applied to compare the LFP power distribution across the seven task epochs ($P < 0.05$, followed by Wilcoxon signed-rank tests to assess directionality of possible modulations (enhancement or suppression, $P < 0.05$ Bonferroni corrected), performed independently for each combination of frequency band (low, 2–8 Hz; medium, 13–28 Hz; high, 60–100 Hz), probe ($n$ = 5), and recording site (32 per probe). For each probe and depth, we compiled a dataset of average LFP power values aggregated across recording sessions, grip types (PG and WH grip), trials (up to 15 per condition). For epochs exhibiting significant (positive or negative) deviation from baseline power, possible differences in LFP power between grip types (e.g., PG vs. WH grip) were tested using paired Wilcoxon signed-rank tests. Sites showing condition-specific effects were logged with corresponding depth, probe, and day identifiers for spatial mapping of task-selective modulations.

To assess representational similarity of LFP activity between agents (i.e. monkey and experimenter) (Fig. 3f), we computed site-wise Pearson correlation coefficients for time-frequency power between the compared conditions. For each frequency ($f$) at the same site ($i$), correlation coefficient ($\rho_{(f,i)}$) was computed by correlating the time-frequency power values from the two conditions:

$$\rho_{(f,i)} = corr\left(P_{cond1(f_i,:)}, P_{cond2(f_i,:)}\right) \tag{3}$$

where $P_{cond1(f_i,:)}$ and $P_{cond2(f_i,:)}$ represent the power over time associated to the two conditions. The resulting correlation values were aggregated (100 bins, range −1 to 1) to visualize the distribution of condition-specific similarity across the frequency spectrum. This resulted in a frequency-by-correlation heatmap, where color intensity reflects the proportion of recording sites exhibiting a given level of similarity. The distribution of $\rho_f$ values across sites was tested for significance using one-sample t-tests.

To evaluate possible differences in LFP modulation across depths (Supplementary Fig. 2a), average power within the three frequency bands was computed during the reaching epoch and aggregated per site. Regression analysis was performed by fitting:

$$y = \beta_0 + \beta_1 * d \tag{4}$$

where $y$ is normalized power and $d$ is the site depth. To quantify spatial gradients in task-related activity, we extracted the slope ($\beta_1$), the coefficient of determination ($R^2$), and the Spearman correlation coefficient ($\rho$).

Day-to-day temporal consistency of LFP modulations (Supplementary Fig. 2b) was assessed in a similar manner to depth-related analysis: normalized power was averaged across sites for each configuration and recording day, and modulation trends over time were evaluated using linear regression and correlation metrics.

### Single-unit analyses

Single units were identified using dedicated offline sorting software (Offline Sorter™ by Plexon Inc) by imposing a 3-standard-deviation negative threshold relative to the signal-to-noise ratio on the band-pass filtered signal (4 pole Bessel filter, 300–7000 Hz) for waveform detection. The temporal stability of each isolated unit throughout the task was verified by projecting spike waveforms into a 3D space defined by the first two principal components and the acquisition time across the entire session. We excluded from our analyses only unstable or non-physiologically plausible waveforms, but not those belonging to units with very low spike count.

Recordings obtained from the same implanted probe across multiple sessions on different days were compared to determine whether and when waveforms with the same spike shape, interspike interval, and response profile during the task could be detected across

multiple days from the same (or adjacent) recording site. Next, to avoid resampling biases, neurons consistently identified in the same or adjacent channels based on these criteria were considered only once in the data set, using as a criterion the signal-to-noise ratio (i.e. the amplitude of the averaged waveform for the unit relative to the threshold of each session).

Cue-related activity was assessed using a $2 \times 3$ repeated-measures ANOVA (factors: Grip and Epoch) separately for each agent condition, possibly followed by post-hoc tests ($p < 0.01$, Tuckey-corrected). Action-related responses during the monkey's own and/or the experimenter's action were assessed with a $2 \times 4$ repeated-measures ANOVA (factors: Grip and Epoch), possibly followed by post-hoc tests ($p < 0.01$, Tuckey corrected). The sign of the modulation of each recorded neurons was assessed relative to its mean baseline firing rate to characterize significant modulations as either positive (facilitated neurons) or negative (suppressed neurons). All neurons showing a significant main effect of the factor Grip or an interaction between Grip and Epoch ($p < 0.01$, Tuckey corrected), were considered as grip-selective action-related neurons. For each neuron, the best grip was defined as the grip type associated with the mean firing rate across the significantly modulated motor epochs (i.e. reaching and grasping) that deviated the most, in absolute terms, from the neuron's baseline firing rate. Action-related neurons were further distinguished based on their agent selectivity as: *self-type* (ST), if their activity differed significantly from the baseline exclusively during monkeys' own trials in the dark condition; *other-type* (OT), if they responded significantly only during the experimenter's trials in the light; or *self-and-other-type* (SOT), if they responded significantly during both conditions. All other comparisons were carried out with ad-hoc ANOVAs with the same statistical thresholds.

## Heat maps and population analyses
Heat maps (Figs. 4b, 4c, 6d) were generated to illustrate the temporal activation profile of individual neurons within selected subpopulations. Each row represents the net soft-normalized activity of a single unit, averaged across 30 trials ($n = 15$ for each grip type), shown separately for self- and other-trials. Soft normalization was performed by subtracting the baseline firing rate from the binned firing rate of each neuron and then dividing the resulting values by the absolute maximum across all conditions for that neuron plus a constant (5 spikes/s), to stabilize variance and reduce the influence of low firing rate units. All final plots were generated using a bin size of 200 ms and a step size of 20 ms.

To examine population activity during the trial (Fig. 3h), we assessed neuronal selectivity between pairwise comparison of the two levels of each investigated factor at successive time bins using a two-tailed sliding $t$-test ($\alpha = 0.01$).

Population plots (Figs. 4–6) were obtained by averaging the single-neuron activity used for the heat maps in 200-ms bins, stepped every 20 ms.

## Preference indices
To quantitatively assess the degree of preference expressed by single neurons for the grip type during self and other trials, a preference index (PI) was calculated as follows:

$$PI = \frac{(R_{PG} - R_{WH})}{(R_{PG} + R_{WH})} \qquad (5)$$

where $R_{PG}$ and $R_{WH}$ are the average responses of the neuron in the reaching/grasping epochs of PG and WH conditions, respectively. The PI values range from 1 (complete selectivity for PG condition) to −1 (complete selectivity for WH), and a value of zero corresponds to identical discharges in the two conditions.

## Correlation analyses
Correlation analyses were performed by means of a two-tailed Pearson's correlation test, carried out on peak of activity timing and discharge onset. The peak time was calculated on the same activity vector computed for the heatmap and defined as the bin with the highest (or lowest, for suppressed neurons) value of mean firing rate across the 15 trials of each type of grip. The discharge onset was defined as the first of a series of consecutive bins up to the peak of activity in the preferred grip condition that were significantly different from the baseline (one-tailed $t$-test, $p < 0.05$).

## Decoding analyses
We employed the Neural Decoding Toolbox[64] to measure the decoding accuracy of a Poisson naive Bayes classifier trained and tested to discriminate self- from other's action using different sets of neurons. For each neuron, data were first converted from raster format into binned format. Specifically, we created binned data that contained the average firing rate in 200-ms bins sampled at 20 ms intervals for each trial (data point). We obtained a population of binned data characterized by a number of data points corresponding to the number of trials per conditions (i.e. $30 \times 2 = 60$ data points for self/other decoding) in an N-dimensional space (where N is the total number of neurons considered for each analysis). Next, we randomly grouped all the available data points into a number of splits corresponding to the number of data points per condition, with each split containing a "pseudo-population", that is, a population of neurons that could be partially recorded separately but treated as if they were recorded simultaneously. Before sending the data to the classifier, we pre-selected those features (neurons) that showed a difference between conditions with $p < 0.5$. The classifier was trained using a leave-one-split-out cross-validation procedure (30 folds for self-other decoding), with accuracy defined as the proportion of correctly classified trials during testing.

## Tracers' injections and histological procedures
At the end of the recordings, Mk2 underwent surgery, as described above, for the injection of retrograde neural tracers into the putaminal zones where the last neural activity had been recorded. Tracers were slowly pressure-injected through a stainless steel, 31-gauge beveled needle connected via a polyethylene tube to a Hamilton syringe (Reno, NV, USA). The needle tip was lowered 4.5 mm below the end of the implanted guide tubes. At the level of the medial guide tube, we injected Fast Blue (FB, 3% in distilled water, Drilling Plastics GmbH, Breuberg, Germany), whereas at the level of the lateral guide tube we injected cholera toxin B subunit, conjugated with Alexa Fluor™ 488 (CTB-g, 1% in phosphate-buffered saline; Molecular Probes).

After 28 days of survival period for tracers' transport in Mk2 and after the end of the experiments in Mk1, the animals underwent deep anaesthesia induced by ketamine and medetomidine, followed by a lethal dose of sodium thiopental. Subsequently, they were perfused through the left cardiac ventricle in sequential stages with approximately 2 litres of saline (over 10 min), 5 litres of 3.5% formaldehyde (over 30 min), and 3 litres of 5% glycerol (over 20 min). All perfusion solutions were prepared in 0.1 M phosphate buffer at pH 7.4.

The brains were then coronally blocked on a stereotaxic apparatus, extracted from the skulls, and placed in 10% buffered glycerol for 3 days, followed by 20% buffered glycerol for 4 days. Finally, the brains were frozen and cut into coronal sections of 60 mm of thickness and collected in five series (sections spaced 300 μm apart). In Mk2, in which FB and CTBg were injected, one series of each fifth section was processed for visualizing CTBg with immunohistochemistry, as described in previous studies[16]. In both Mk1 and Mk2, two series were stained using the Nissl method (0.1 thionin in 0.1 M acetate buffer at pH 3.7).

The distribution of FB and CTBg labelled neurons in the cortex of Mk2 was plotted in sections every 600 μm together with the outer and inner cortical borders, using a computer-based charting system. The location of the electrodes' tracks in Mk1 and Mk2 was verified in Nissl-stained sections and then plotted in sections every 600 μm with the outer border of the striatum. Data from individual sections were then imported into the 3D reconstruction software to obtain volumetric reconstruction of the injected hemisphere and of the recorded striatum. The areal attribution of the labelled neurons was based on criteria and maps described in previous studies[65]. The cortical input to the injected striatal zone was expressed in terms of the percentage of labelled neurons found in each cortical area with respect to the overall cortical labelling found for each tracer's injection in the injected hemisphere.

### Reporting summary

Further information on research design is available in the Nature Portfolio Reporting Summary linked to this article.

## Data availability

The source data are provided in the Source Data file available at: https://doi.org/10.17605/OSF.IO/82JFR. The raw electrophysiological datasets generated in this study include additional measurements that are part of ongoing work: the full dataset is available from the corresponding author upon request. Source data are provided with this paper.

## Code availability

No custom code was generated in this study. Data analyses were conducted using standard, built-in MATLAB functions.

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

## Acknowledgements

We thank F. Mastrandrea for drawings; A. Guida for animal care and technical assistance; G. Luppino for technical assistance; F. Leonardi for veterinary assistance and anesthesia; A. Mitola, E. Arcuri, A. Camisa and A. Mancuso for help in animal handling and data acquisition. European Research Council (ERC StG-2015) grant WIRELESS (678307), ERC CoG-2020 grant EMACTIVE (101002704), ERC PoC-2020 grant FUTURE-NHP (957538) to LB; NEXTGENERATIONEU project MNESYS (PE0000006–DN. 1553 11.10.2022) to LB; Italian Ministry of University and Research (MUR) Grant FARE2020, project n. R20NJ7BBA7 "CIRCEM"; MUR Grant FARE2017, project n. R16PWSFBPL "GANGLIA" to LB; MUR Grant PRIN 2022 n. 22SP5K99 to MM.

## Author contributions

Conceptualization: L.B. Design of the experiments: C.R., M.M., L.B. Animal preparation and data acquisition: C.R., M.R., C.G.F., M.M., L.B. Data analysis and visualization: C.R., M.R., E.I., G.B., and E.B. Supervision: L.B. and M.M. Funding acquisition: L.B. and M.M. Project administration: L.B. Writing—original draft: C.R. and L.B. Writing—review and editing: C.R., L.B., M.M., M.R., C.G.F., E.I., G.B., and E.B.

## Competing interests

The authors declare no competing interests.
