## [Transparent Peer Review file · Nature Communications]

Pragmatic representations of self and others' action in the monkey putamen

Corresponding Author: Professor Luca Bonini

Version 0:

Reviewer comments:

Reviewer #1

(Remarks to the Author)

This manuscript describes a study of primate putamen neurons in a motor task involving execution and observation of different actions under different conditions. The design varied agent (self, other) and grip type (precision, full-hand), instructed by auditory and visual cues, as well as performance under observation vs in-darkness conditions. The monkey was rewarded on all (self, other) correct trials. LFP low/medium frequency-band signals, likely reflecting cortical input, depended on contextual information provided by the audio-visual cues, with higher selectivity on self compared to other trials. The majority of putamen single-neurons did not show cue-related activity but activity related to the motor responses, with prominent population coding only of self-vs-other and self grip type. Most neurons were selective for self actions although meaningful numbers of neurons also encoded either only other's actions or both self and other's actions. These neurons showed differential sensitivity to visible vs non-visible context and whether or not a barrier was introduced.

Overall, these are interesting and well-conducted experiments. The manuscript is well-written and clear. Analyses are appropriate. An unusual feature of the study is the connectivity characterization using retrograde tracing from the putamen recording site, which provides strong support for the characterization of the recorded neurons as being part of a motor-coordination network. The manuscript by itself does not necessarily offer a strong mechanistic advance. However, its value and importance lie, in my view, in introducing to the literature rare and precious evidence on the response properties of primate putamen neurons. This is important not only for advancing understanding of the primate including human functional networks for action execution and social action-coordination, but also relevant from a clinical perspective given the basal ganglia's role in various neurological diseases. The data will be of interest also to computational neuroscientists who critically require such data to build more refined theories of neural motor control systems. A few comments should be addressed before publication.

Major points

1. One concern of a potential confound, or at least an alternative explanation, for the authors' interpretation of a 'pragmatic' representation is whether the activity of putamen neurons reflects covert muscle activation. This interpretation would explain why the neurons do not respond when a barrier is introduced. This could be assessed by recordings using muscle electrodes. Although it may not be possible to conduct these control tests (which should not preclude publication of this interesting data set), this issue should at least be discussed.
2. It would be helpful if the authors could provide evidence to confirm that the monkeys actually observed the partner's actions, e.g. eye-fixation data. If no such data is available, it would be important to discuss the issue, i.e. whether differences in putamen activity on self and other trials might be related to differences in visual stimulation or attention.
3. Perhaps the most puzzling finding is that activity of neurons responding to other's action did not reflect whether or not the action was visible. On the one hand, this result is consistent with the absence of visibility-related modulation in own-action coding neurons. However, it remains unclear whether the monkeys actually processed the other's action. The authors should discuss whether additional controls might be necessary to validate this finding, e.g. a task in which the monkey receives reward only if it correctly responds to the other's action. An alternative explanation that should be discussed is also that putamen neurons coding other's action in the dark might not be motor related but instead provide a timing signal of an event that predicts reward. This could be tested by recording neurons 'during extinction', i.e. when other's action is no longer

rewarded.

4. Related to the above point, it would be important to test whether some neurons respond selectively during reward period. More specifically, do neurons respond differentially based on whether reward results from own vs other's action? This is of considerable interest as a previous study showed agency-dependent reward coding in the monkey striatum (Baez-Mendoza, Harris & Schultz, 2013, PNAS).

Minor points

5. Please define pragmatic on first use, i.e. as it relates to a neural representation and also clarify this in line 315.

6. The introduction should refer to classical primate neurophysiology studies on the putamen, e.g., by Kimura, Romo.

7. Typo line 191: thee

Reviewer #2

(Remarks to the Author)

In the current study, the authors investigated neural coding of self vs. other hand motor behavior in putamen. They introduced three experimental conditions: self vs. other, precision grip vs. whole hand, light vs. dark. The main finding is that putamen neurons encode the agent, grip type, but not the light variable. For self-action, putamen encodes specific grip type, but not the agent. These findings extend previous research of cortical neurons, with some of them consistent with those in cortex, but there are also difference. Thus overall this study has novelty, and extend our understandings of hand action coding from cortex to subcortical areas. The experiments are nicely designed. The paper is clearly written.

Major comments:

1. The other-action neurons do not distinguish between the light and dark condition, which is a bit puzzling and unexpected to me. As the author pointed out, this result may imply that the animals infer or imagine that the experimenter performs the task, without really "seeing" it. A control could be something like the experimenter hold their hand without reaching the task although he/she needs to perform reach during that trial. For example, similar to their last control experiment, the transparent obstacle could be placed between the experimental and the target. Theoretically the neuron will not respond in this case, isn't it?
2. Related to the first point, it is also not expected to me that the other-action neurons do not encode the grip type. I thought these neurons should encode something shared between self and other action, if they are mirror neurons, so that they should distinguish precision grip or whole hand for the experimenters. Yet this is not the case. What does this finding imply? The author could provide more insight and discussion.
3. I have difficult to understand the logic of the LFP result. If the LFP reflects input from cortex to putamen, then the context information about agent reflected in the LFP, implying encoded in cortex, is also expected in putamen (spiking activity). Yet this information is not reflected in the spiking activity of putamen. So where does this context information go? Does it go to elsewhere?

Minor comments:

1. Reverse figure sequence: Figure 1 behavior; Figure 2 Putamen recording?
2. Figure 2, figures are a bit small, better incorporate the supplementary figure 2 about the two types of grip
3. I suggest to have a summary figure (e.g. box and arrows), showing possible functional difference between cortical areas and putamen, based on their current findings. This would help readers to visualize their conclusions and hypothesis.

Reviewer #3

(Remarks to the Author)

Rotunno and colleagues carefully designed a well-controlled MAT task to examine the neural representation of self and others' actions. It is striking that spiking activity and LFPs in the putamen—an area anatomically close to the fronto-parietal network—reflect others' actions in a manner similar to self-actions, particularly when the animal can potentially interact with the other. I was convinced that putamen neurons were modulated by multiple task-related factors. However, the distinct functional roles of the putamen compared to cortical regions during the MAT task remain somewhat unclear. The quality of the collected data is high, and I believe these original findings will attract considerable interest in the field of social neuroscience. I hope the following suggestions may help strengthen the manuscript.

1. The behavioral results in Fig. 2 are robust, clearly showing that the animals learned the task rule, with consistently low error rates across individuals. However, how did performance differ between self-action and other-action trials? Could differences in neuronal firing between self and other trials be partly explained by task difficulty or cognitive demand?
2. To better address the role of the putamen and associated cortical areas in task performance, it would be informative to focus on error trials. How did spiking activity and LFPs differ between correct and error trials?
3. The authors convincingly show that anatomical projections from the fronto-parietal network differ between medial and

lateral putamen. It is striking that medial putamen, but not lateral putamen, receives dominant projections from F7, a key mirror neuron area. To further support the claim that others' action representations in putamen arise from cortical connectivity, I suggest directly comparing spiking and LFP patterns between medial and lateral recording sites. Comparing effects across Figures 3–6 (e.g., the counts of cell numbers based on response selectivity, mean PSTHs, etc.) between medial and lateral putamen would considerably strengthen the argument.

4. The effect of the barrier on representations of others' actions in putamen activity is especially intriguing. Related to the above point, how were barrier-sensitive neurons distributed across the putamen? Were they concentrated in medial regions?

5. The frequency distributions of the grip Preference Index during self- versus other-action trials are notably different. How do the authors interpret this divergence in putamen cell activity? Is the result consistent with findings from cortical recordings within the fronto-parietal network? Further discussion would be valuable.

Version 1:

Reviewer comments:

Reviewer #1

(Remarks to the Author)

The authors have addressed all my comments.

Reviewer #2

(Remarks to the Author)

The authors have addressed my questions well. I don't insist with my some of my previous minor suggestions. Supporting for publication.

Reviewer #3

(Remarks to the Author)

I appreciate the authors' responses to my comments.

Regarding your response to my third comment, I understand that the projection from F7 is relatively minor in both the medial and lateral putamen. The authors compared neuronal activity profiles between these two regions but could not draw any firm conclusion about meaningful differences worth including in the manuscript.

If that is the case, presenting the "differential projection data between the medial and lateral putamen" may be misleading in supporting the electrophysiological findings. For publication, I recommend either omitting these data altogether or focusing solely on one subregion—medial or lateral putamen—to demonstrate that the putamen, as a whole, is connected with broad motor control cortical regions.

Regarding your response to my fifth comment, I am convinced that the mismatch between self and other's grip coding also occurs in some cortical regions. However, it remains unclear whether the pattern of inconsistency is consistent between the cortex and the putamen. My point differs from that raised in comment 2 by Reviewer 2. I suggest that examining the consistency of this "self–other inconsistency pattern" between the cortex and putamen could provide strong evidence as to whether the same information is represented in both structures or whether some transformation occurs.

The authors have not fully addressed this issue, and I recommend that they discuss it in greater detail in the Discussion section.

We sincerely thank all the Reviewers for appreciating and understanding the novelty and relevance of our work, as well as for the constructive comments, which we have addressed in the revised version of the manuscript.

We believe that these revisions significantly enhanced the clarity and robustness of the paper. Point-by-point responses to each Reviewer's comment are provided below, and the main changes/addition to the text are highlighted in red font in the revised manuscript.

REVIEWER COMMENTS

Reviewer #1 (Remarks to the Author):

This manuscript describes a study of primate putamen neurons in a motor task involving execution and observation of different actions under different conditions. The design varied agent (self, other) and grip type (precision, full-hand), instructed by auditory and visual cues, as well as performance under observation vs in-darkness conditions. The monkey was rewarded on all (self, other) correct trials. LFP low/medium frequency-band signals, likely reflecting cortical input, depended on contextual information provided by the audio-visual cues, with higher selectivity on self compared to other trials. The majority of putamen single-neurons did not show cue-related activity but activity related to the motor responses, with prominent population coding only of self-vs-other and self grip type. Most neurons were selective for self actions although meaningful numbers of neurons also encoded either only other's actions or both self and other's actions. These neurons showed differential sensitivity to visible vs non-visible context and whether or not a barrier was introduced. Overall, these are interesting and well-conducted experiments. The manuscript is well-written and clear. Analyses are appropriate. An unusual feature of the study is the connectivity characterization using retrograde tracing from the putamen recording site, which provides strong support for the characterization of the recorded neurons as being part of a motor-coordination network. The manuscript by itself does not necessarily offer a strong mechanistic advance. However, its value and importance lie, in my view, in introducing to the literature rare and precious evidence on the response properties of primate putamen neurons. This is important not only for advancing understanding of the primate including human functional networks for action execution and social action-coordination, but also relevant from a clinical perspective given the basal ganglia's role in various neurological diseases. The data will be of interest also to computational neuroscientists who critically require such data to build more refined theories of neural motor control systems. A few comments should be addressed before publication.

R. We deeply thank the Reviewer for having recognized and emphasized all the main advancements and strengths of our work.

Major points

1. One concern of a potential confound, or at least an alternative explanation, for the authors' interpretation of a 'pragmatic' representation is whether the activity of putamen neurons reflects covert muscle activation. This interpretation would explain why the neurons do not respond when a barrier is introduced. This could be assessed by recordings using muscle electrodes. Although it may not be possible to conduct these control tests (which should not preclude publication of this interesting data set), this issue should at least be discussed.

R. Recording muscle activity is not possible since both animals have been euthanized to obtain anatomical reconstruction. We did not record EMG because in previous works of our (Bonini et al. 2014 J Neurosci; Maranesi et al. 2015 J Neurosci; Livi et al. 2019 PNAS) and others' groups (Kraskov et al. 2009) it was repeatedly established that the presence of the experimenter that performs the task instead of the monkey in specific and predictable trials acts, as a matter of fact, as a "social barrier" that inhibits the monkey from moving and prevents EMG activity to occur. In line with this literature, we added in the revised manuscript a Supplementary Figure (Supplementary Figure 4) where we show that self-type neurons - which are active during monkey's own action, when EMG activity is obviously high – remain silent during experimenter's action, when simultaneously recorded other-related neurons discharge. Although indirectly, this evidence suggests that the activity of other-related putamen neurons is hardly accountable for subtle muscle activation.

All these aspects, the reference to the new Supplementary figure, and the relevant literature, have been included in the Results section titled “Putamen neurons encoding self and other’s action” and commented in a paragraph dedicated to possible alternative interpretations at the end of the Discussion section (see text in red font).

2. It would be helpful if the authors could provide evidence to confirm that the monkeys actually observed the partner’s actions, e.g. eye-fixation data. If no such data is available, it would be important to discuss the issue, i.e. whether differences in putamen activity on self and other trials might be related to differences in visual stimulation or attention.

R. This is an important point that we carefully considered before starting the experiments. Since we wanted to keep the behavior as natural as possible, avoiding restraining the monkeys’ head (see Lanzarini et al. 2025 Science) and leaving visual exploration completely free, we couldn’t measure eye position during the task. Of course, the monkey was required to obtain information from the contextual cues to correctly perform the task, which implied paying attention to the workspace shared with the experimenter. Nevertheless, there might be differences in the amount of gaze toward the workspace during self- and other-trials. Previous data from our (Maranesi et al. 2013 Eur J Neurosci) and other (Papadourakis and Raos 2019 Cereb Cortex) groups, showed that gaze dependence of action observation responses in the premotor cortex (which is a strong source of projections to the putamen) is generally low. Furthermore, neurophysiological experiments on action observation that used free-gaze paradigms reported the highest visual responses and selectivity for object/grip features (Mazurek et al. 2018 J Neurosci; Papadourakis and Raos 2019 Cereb Cortex). Thus, free visual exploration seems to be an important condition to maximize the recruitment of motor brain regions during action observation, and this is why we decided to perform the experiments using this approach.

Concerning the possible differences in putamen neuron activity between self and other trials, it is unlikely that they could be due to visual stimulation or attention because other-related neurons responded even during trials in the dark. We elaborated more on these issues in the dedicated paragraph at the end of the Discussion section (see text in red font).

3. Perhaps the most puzzling finding is that activity of neurons responding to other’s action did not reflect whether or not the action was visible. On the one hand, this result is consistent with the absence of visibility-related modulation in own-action coding neurons. However, it remains unclear whether the monkeys actually processed the other’s action. The authors should discuss whether additional controls might be necessary to validate this finding, e.g. a task in which the monkey receives reward only if it correctly responds to the other’s action. An alternative explanation that should be discussed is also that putamen neurons coding other’s action in the dark might not be motor related but instead provide a timing signal of an event that predicts reward. This could be tested by recording neurons ‘during extinction’, i.e. when other’s action is no longer rewarded.

R. Concerning the first point, it should be noted that the monkey was required to remain still during experimenter’s turns to receive the reward, which was a constant of all correctly performed trials. This makes absolutely reasonable the Reviewer’s alternative explanation: since the reward was a constant of all self and other correctly performed trials, one would expect that reward-prediction neurons would fire similarly during both agents’ trials. Indeed, SOT neurons exhibit some similarities in their peak response time and discharge onset (see Fig. 4e), which would fit with the interpretation proposed by the Reviewer. However, if neuronal discharge was a timing rather than a motor signal, these similarities should be maintained also when comparing other and barrier conditions, as the timing of the reward did not vary between them. The lack of response in the barrier condition, however, strikingly contrasts with this interpretation, bringing support to the pragmatic hypothesis: we added these considerations at the end of the Discussion section (see text in red font) and we thank the Reviewer for pointing this out.

4. Related to the above point, it would be important to test whether some neurons respond selectively during reward period. More specifically, do neurons respond differentially based on whether reward results from own vs other’s action? This is of considerable interest as a previous study showed agency-dependent reward coding in the monkey striatum (Baez-Mendoza, Harris & Schultz, 2013, PNAS).

R. We thank the Reviewer for raising this very interesting point. In the present manuscript, we deliberately excluded from the analysis the reward epoch in order to avoid confounding reward-related responses with task-related activity, as specified in the Methods section. Nonetheless, we totally agree that this issue is of considerable interest, thus we followed the Reviewer’s suggestion conducting a preliminary analysis (2 x 3 repeated measures ANOVA, factors: Epoch and Condition), focused on a comparison of baseline and reward epochs (500 ms epoch following reward delivery, as in Baez-Mendoza et al. 2013) during self-, partner-social, and partner-barrier conditions. This analysis was applied to all neurons in the dataset, including the non-task-related ones because these latter could respond to reward delivery as this epoch was excluded from the main analyses. The following figure summarizes the main findings.

It is worth noting that both task-related (52%) and task-unrelated (34%) neurons exhibit possible reward-related responses. In addition, as predicted by the Reviewer, most of the neurons appear to show agency-dependence as in Baez-Mendoza’s paper. Indeed, possible mouth-related motor responses may occur during reward delivery, but they should be present in all the three conditions: this never occurred, as illustrated in the figure above, thereby supporting the idea of agent-dependent reward coding.

However, reward coding through the lens of our task is a more complex phenomenon. For example, the activity of at least part of the neurons responding selectively during self-trials may actually be associated with the arm returning to the button after task execution (an action that could evoke an arm-related motor response even in task-unrelated neurons) rather than to the reward delivery, which occurs simultaneously. In addition, we found that most of the neurons encoding the reward following other-trials exhibited specificity for the social or barrier condition, suggesting a more complex, context-dependent form of agent-based reward coding in the putamen nucleus.

We deeply thank the Reviewer for the great suggestion, and given the interest, richness, and complexity of these findings we decided to further investigate this important point and publish it in a dedicated paper.

Minor points

Please define pragmatic on first use, i.e. as it relates to a neural representation and also clarify this in line 315.

R. We did this on the first occurrence of the term in the introduction, specifying that the pragmatic nature of visual responses pertains to “their role in affording object-directed actions as well as motor responses to others during social interactions”.

6. The introduction should refer to classical primate neurophysiology studies on the putamen, e.g., by Kimura, Romo.

R. Done, thanks for pointing this out.

7. Typo line 191: thee.

R. Done.

Reviewer #2 (Remarks to the Author):

In the current study, the authors investigated neural coding of self vs. other hand motor behavior in putamen. They introduced three experimental conditions: self vs. other, precision grip vs. whole hand, light vs. dark. The main finding is that putamen neurons encode the agent, grip type, but not the light variable. For self-action, putamen encodes specific grip type, but not the agent. These findings extend previous research of cortical neurons, with some of them consistent with those in cortex, but there are also difference. Thus overall this study has novelty, and extend our understandings of hand action coding from cortex to subcortical areas. The experiments are nicely designed. The paper is clearly written.

R. We deeply thank this Reviewer for recognizing the novelty of our work, the accuracy of the experimental design, and the care we devoted to the writing.

Major comments:

1. The other-action neurons do not distinguish between the light and dark condition, which is a bit puzzling and unexpected to me. As the author pointed out, this result may imply that the animals infer or imagine that the experimenter performs the task, without really “seeing” it. A control could be something like the experimenter hold their hand without reaching the task although he/she needs to perform reach during that trial. For example, similar to their last control experiment, the transparent obstacle could be placed between the experimental and the target. Theoretically the neuron will not respond in this case, isn't it?

R. Regarding the first point, we did not have a condition in which the experimenter voluntarily omits to act, but it could have been interesting to analyze the response during erroneous omitted experimenter's action (we did it for omitted monkey's action in a previous study, see Bonini et al. 2014 Curr Biol); however, it is not feasible with the current data set because of the small number of total errors of both agents, which prevents us from having sufficient statistical power (see also our Response to point 2 of Reviewer 3).

Concerning the second point, we did not test the condition in which a barrier was interposed between the experimenter and the target. However, based on our interpretation of the generative properties of the putamen neurons - derived from previous evidence from other regions of the cortical grasping network - it should not be surprising to observe agent-selective neuronal responses even in the absence of anyone's movement, since this has been reported in the ventral premotor cortex (see for example Bonini et al. 2014, Curr Biol), a powerful source of input to the putamen.

Although putamen neurons' responses during other trials could have alternative interpretations (as also pointed out by Reviewer 1), these interpretation do not appear to fit with our findings (see our responses to points 3 and 4 of Reviewer 1), suggesting that OT and SOT neurons' response during other trials cannot be explained in terms of reward prediction or task staging but, most likely, reflects a pragmatic representation of action.

2. Related to the first point, it is also not expected to me that the other-action neurons do not encode the grip type. I thought these neurons should encode something shared between self and other action, if they are mirror neurons, so that they should distinguish precision grip or whole hand for the experimenters. Yet this is not the case. What does this finding imply? The author could provide more insight and discussion.

R. In line with our recent theoretical contributions (Orban et al. 2021; TiCS; Bonini et al. 2022; 2023 TiCS), we do not use the term “mirror” in the paper. We strongly believe that most cortical cells encoding others' actions are better understood within a broader, dynamic “other-to-self” framework, rather than as a simple and rigid “one-to-one matching” between self and other actions evoked by the original “mirror” concept.

Regarding grip tuning, we actually show that a few other-related neurons discharge differently during the experimenter's grip types, but their tuning is completely mismatched between self and other actions.

This finding is not at odds with many reports on cortical “mirror” neurons which – when properly tested – reveal no match between self and other grip-type coding (Pani et al. 2014 J Cogn Neurosci; Papadourakis and Raos 2019 Cereb Cortex; Maranesi et al. 2024 Prog Neurobiol). We stated this explicitly in the Discussion and we have further elaborated this concept in the revised version.

3. I have difficulty to understand the logic of the LFP result. If the LFP reflects input from cortex to putamen, then the context information about agent reflected in the LFP, implying encoded in cortex, is also expected in putamen (spiking activity). Yet this information is not reflected in the spiking activity of putamen. So where does this context information go? Does it go to elsewhere?

R. We agree that the presence of context-related information in the LFP but not in the spiking activity may at first seem counterintuitive. However, as extensively shown by existing literature (see Buzsáki et al. 2012, Nat Rev Neurosci) even in primate motor cortex (Confais et al., 2020; Rule et al. 2017) LFPs and spikes reflect different aspects of neuronal processing: specifically, LFPs primarily capture the aggregate synaptic and dendritic currents that constitute the facilitatory and inhibitory input to a local network, which not necessarily turn into changes in spiking activity of individual neurons.

As such, input information may be clearly represented in the LFP signal, as it is driven by spiking activity of other areas but not necessarily reflected in local spiking activity of the target area. This local filtering may be supported by inhibitory interneurons and local circuitries that we have hypothesized and considered in different sections of the paper (e.g. “*while self-type neuronal responses are likely driven by direct cortico-striatal projections*”³⁵, *other-type responses may be more strongly shaped by local inhibitory mechanisms*” in the Results section, or “*extends to the basal ganglia the possible function to modulate the decision of whether or not to respond to the action of another agent, while delegating to the cortico-cortical sensorimotor circuit the specification of the details (such as the grip type) concerning how to do so, based on object- or agent-specific affordances*” in the Discussion).

Minor comments:

1. Reverse figure sequence: Figure 1 behavior; Figure 2 Putamen recording?

R. We prefer to keep the original sequence of the figures as it also reflects the scientific path we followed, from previous neuroanatomical evidence lacking a physiological counterpart to the current functional investigation of anatomically-characterized regions.

2. Figure 2, figures are a bit small, better incorporate the supplementary figure 2 about the two types of grip.

R. We did it, thanks for the suggestion.

3. I suggest to have a summary figure (e.g. box and arrows), showing possible functional difference between cortical areas and putamen, based on their current findings. This would help readers to visualize their conclusions and hypothesis.

R. We totally agree with the Reviewer, but similar figures have already been presented in previous review articles in which hypotheses tested in this work were advanced (see Figure 1 in Bonini 2017 Neuroscientist and Figure 3 in Bonini et al. 2022 TiCS). At this stage, we believe that the available data provides a significant advancement in the understanding of the functional properties of putamen neurons, but more work is still needed to advance our understanding of the relationship between cortical areas and the putamen.

Reviewer #3 (Remarks to the Author):

Rotunno and colleagues carefully designed a well-controlled MAT task to examine the neural representation of self and others' actions. It is striking that spiking activity and LFPs in the putamen—an area anatomically close to the fronto-parietal network—reflect others' actions in a manner similar to self-actions, particularly when the animal can potentially interact with the other. I was convinced that putamen neurons were modulated by multiple task-related factors. However, the distinct functional roles of the putamen compared to cortical regions during the MAT task remain somewhat unclear. The quality of the collected data is high, and I believe these original findings will attract considerable interest in the field of social neuroscience. I hope the following suggestions may help strengthen the manuscript.

R. We thank the Reviewer for recognizing the quality of our data and for the thoughtful suggestions, which we have carefully addressed as detailed below.

1. The behavioral results in Fig. 2 are robust, clearly showing that the animals learned the task rule, with consistently low error rates across individuals. However, how did performance differ between self-action and other-action trials? Could differences in neuronal firing between self and other trials be partly explained by task difficulty or cognitive demand?

R. We thank the Reviewer for raising this point, which gave us the opportunity to add another category of errors in the behavioral task (*wrong start*) related to erroneous detachment of monkey's hand from the starting position during experimenter's trials. In fact, this type of errors, compared with similar errors during monkey's trials (i.e. early start and no touch, combined), might reveal a greater difficulty for the monkey in inhibiting the tendency to act during experimenter's relative to self-trials, which in turn could explain some differences in neuronal firing between the two types of trials.

However, as shown in the new version of Fig. 2c that integrates the percentage of wrong start during experimenter's trial, in both monkeys the frequency of these errors is similar or even lower during other relative to self trials. These findings do not support the hypothesis that difference in neuronal firing between self and other trials can be explained by higher task difficulty or cognitive demand.

2. To better address the role of the putamen and associated cortical areas in task performance, it would be informative to focus on error trials. How did spiking activity and LFPs differ between correct and error trials?

R. We fully agree that an analysis of error trials would be informative for further clarifying the role of the putamen and associated cortical areas. However, as also noted by the Reviewer in the previous point, both animals made an extremely low number of errors throughout the task. The total number of errors across all sessions is reported in the manuscript, and the distribution of so few trials across sessions does not allow us a meaningful comparison of the activity between correct and error trials.

3. The authors convincingly show that anatomical projections from the fronto-parietal network differ between medial and lateral putamen. It is striking that medial putamen, but not lateral putamen, receives dominant projections from F7, a key mirror neuron area. To further support the claim that others' action representations in putamen arise from cortical connectivity, I suggest directly comparing spiking and LFP patterns between medial and lateral recording sites. Comparing effects across Figures 3–6 (e.g., the counts of cell numbers based on response selectivity, mean PSTHs, etc.) between medial and lateral putamen would considerably strengthen the argument.

R. We agree that anatomical projections from the fronto-parietal network show partially distinct topographies between medial and lateral putamen. However, our tracer data indicate that the contribution of F7 is in fact very limited in both cases, while the majority of premotor labeling involved other areas such as F5, F3 and F2. Following FB injection, F7 labeling appeared more clustered and accounted for about 0.8% of the total labeling; by contrast, after CTBg injection, labeling was more

scattered and did not reach the 0.5% threshold used to display areas in panel e. We thank the Reviewer for pointing this out: to avoid any confusion, we have now specified this in the legend of Fig. 1.

Despite the small quantitative differences observed in tracer injections, we followed the Reviewer's suggestion and directly compared the distribution of different types of neurons based on their response type between medial and lateral probes, in all implants of both monkeys. Chi-square tests on the relative frequencies of ST, SOT and OT neurons showed no significant difference between medial and lateral probes (ST: $\chi^2 = 0.0163$, $p = 0.9$; SOT: $\chi^2 = 0.108$, $p = 0.74$; OT: $\chi^2 = 0.064$, $p = 0.80$). Furthermore, since we used linear probes for neuronal recordings, we also compared possible inhomogeneity in the depth distribution of ST, SOT and OT neurons between dorsal, intermediate and ventral recording sites. Chi-square tests revealed an overall homogeneity in ST ($\chi^2 = 2.69$, $p = 0.26$) and SOT neurons ($\chi^2 = 0.90$, $p = 0.636$) distributions across different depths. On the contrary, OT neurons showed an under-representation at dorsal positions and an over-representation at intermediate sites ($\chi^2 = 19.40$, $p < 0.001$). The figure below illustrates these findings.

Depth distribution of task-related neurons. **a,b**, Percentage distributions of ST (light blue), OT (green), and SOT (yellow) and barrier-related (coral) neurons across dorsal, intermediate and ventral sites. Chi-square tests revealed no significant differences in ST ($\chi^2 = 2.69$, $p = 0.26$), SOT ($\chi^2 = 0.90$, $p = 0.636$) and barrier-sensitive ($\chi^2 = 0.64$, $p = 0.73$) distributions across depths, indicating that neuronal classes are homogeneously represented in the dorso-ventral extension of the putamen. By contrast, OT neurons exhibit a significant over-representation at intermediate sites and an under-representation in dorsal sites ($\chi^2 = 19.40$, $p < 0.001$).

We prefer not to include these data in the manuscript because the only significant result regarding OT neurons is likely due to their very low number relative to the others, thus being probably a false positive result (see also the small difference in LFP depth topography in Supplementary Figure 2 and barrier sensitive neurons in the next point).

4. The effect of the barrier on representations of others' actions in putamen activity is especially intriguing. Related to the above point, how were barrier-sensitive neurons distributed across the putamen? Were they concentrated in medial regions?

R. To address this point, we examined whether barrier-sensitive neurons were differentially distributed across the putamen. In medial probes, 14 out of 19 neurons (74%) were classified as barrier-sensitive, whereas in lateral probes 34 out of 51 (67%) showed barrier sensitivity. A chi-square test indicated no significant difference between these proportions ($\chi^2 = 0.099$, $p = 0.75$). As for the previous point (see above), we leveraged our linear probes to test the possible non-uniform depth distribution of barrier-sensitive neurons and did not find any significant inhomogeneity ($\chi^2 = 0.64$, $p = 0.73$). Together, these findings indicate that barrier-sensitive neurons are homogeneously distributed in the putamen nucleus.

5. The frequency distributions of the grip Preference Index during self- versus other-action trials are notably different. How do the authors interpret this divergence in putamen cell activity? Is the result consistent with findings from cortical recordings within the fronto-parietal network? Further discussion would be valuable.

R. We thank this Reviewer that, together with Reviewer 2, raised this point. Indeed, the mismatch between self and other's grip coding in putamen neurons is consistent with findings from cortical recordings and we further elaborated this aspect in the revised Discussion section of the manuscript (please, see our extended response to the same issue at point 2 of Reviewer 2).

We sincerely thank all the reviewers for their endorsement of our manuscript and reviewer 3 for the additional comments, which we have addressed in the revised version as detailed below.

We hope that our paper is now suitable for publication in Nature Communications.

REVIEWER COMMENTS

Reviewer #1 (Remarks to the Author):

The authors have addressed all my comments.

R. We thank the reviewer for the positive feedback.

Reviewer #2 (Remarks to the Author):

The authors have addressed my questions well. I don't insist with my some of my previous minor suggestions. Supporting for publication.

R. We thank the reviewer for the positive feedback.

Reviewer #3 (Remarks to the Author):

I appreciate the authors' responses to my comments.

R. We thank the reviewer for the positive feedback on our responses.

1) Regarding your response to my third comment, I understand that the projection from F7 is relatively minor in both the medial and lateral putamen. The authors compared neuronal activity profiles between these two regions but could not draw any firm conclusion about meaningful differences worth including in the manuscript.

If that is the case, presenting the “differential projection data between the medial and lateral putamen” may be misleading in supporting the electrophysiological findings. For publication, I recommend either omitting these data altogether or focusing solely on one subregion—medial or lateral putamen—to demonstrate that the putamen, as a whole, is connected with broad motor control cortical regions.

R. We fully agree with the reviewer's conclusion that the main purpose of the tracing study is to demonstrate that the putamen, as a whole, is connected with the main nodes of the cortical grasping network. Indeed, despite unavoidable quantitative differences, this is precisely what our data show, and it was the original aim of our analysis. For this reason, we believe it would not be appropriate to omit informative data that further support and strengthen our main conclusion, in line with the reviewer's remark. To further emphasize this point and avoid confusion, we have revised the text in the Results section to remove any reference to comparative descriptions of the tracing data, and we now conclude the section with this sentence: ““*The most lateral part of the putamen exhibits a similar connectivity pattern (Fig. 1d), with strong input from the primary motor and dorsal premotor cortices (Fig. 1e), supporting a prominent role of the putaminal investigated regions in forelimb motor control.*”

2) Regarding your response to my fifth comment, I am convinced that the mismatch between self and other's grip coding also occurs in some cortical regions. However, it remains unclear whether the pattern of inconsistency is consistent between the cortex and the putamen. My point differs from that raised in comment 2 by Reviewer 2. I suggest that examining the consistency of this “self–other inconsistency pattern” between the cortex and putamen could provide strong evidence as to whether the same information is represented in both structures or whether some transformation occurs. The authors have not fully addressed this issue, and I recommend that they discuss it in greater detail in the Discussion section.

R. We thank the reviewer for the clarification regarding the specificity of their point above. It is now clearer what they meant, and we further elaborated this point in the discussion section as follows: ““*The lack of congruence in single-neuron grip tuning between self and other trials is consistent with previous findings in the premotor⁵³ and parietal^{9,54} cortices. However, the proportion of putaminal neurons with visual (10%) and motor (25%) selectivity for the grip type is considerably lower than that reported for all cortical areas for which data are available. Together with our anatomical connectivity data, these findings suggest that within the cortico-striatal system, cortico-cortical sensorimotor circuits specify the detailed features of potential motor plans (e.g., grip type) based on object- or agent-specific affordances, whereas the basal ganglia modulate the decision of whether or not to respond to another agent’s action – and not necessarily with a matched action^{14,55}.*”

Refs:

- [9] Maranesi, M., Lanzilotto, M., Arcuri, E. & Bonini, L. Mixed selectivity in monkey anterior intraparietal area during visual and motor processes. *Prog Neurobiol* **236**, 102611 (2024).
- [14] Bonini, L., Rotunno, C., Arcuri, E. & Gallese, V. The mirror mechanism: linking perception and social interaction. *Trends Cogn Sci* **27**, 220–221 (2023).
- [53] Papadourakis, V. & Raos, V. Neurons in the Macaque Dorsal Premotor Cortex Respond to Execution and Observation of Actions. *Cereb. Cortex* **29**, 4223–4237 (2019).
- [54] Pani, P., Theys, T., Romero, M. C. & Janssen, P. Grasping execution and grasping observation activity of single neurons in the macaque anterior intraparietal area. *J Cogn Neurosci* **26**, 2342–2355 (2014).
- [55] Orban, G. A., Sepe, A. & Bonini, L. Parietal maps of visual signals for bodily action planning. *Brain Struct Funct* **226**, 2967–2988 (2021).